# On non-canonical degrees of freedom

Eoin Quinn

*Université Paris-Saclay, CNRS, LPTMS, 91405 Orsay, France*

epquinn@gmail.com

April 7, 2021

## Abstract

Non-canonical degrees of freedom provide one of the most promising routes towards characterising a range of important phenomena in condensed matter physics. Potential candidates include the pseudogap regime of the cuprates, heavy-fermion behaviour, and also indeed magnetically ordered systems. Nevertheless it remains an open question whether non-canonical algebras can in fact provide legitimate quantum degrees of freedom. In this paper we survey progress made on this topic, complementing distinct approaches so as to obtain a unified description. In particular we obtain a novel exact representation for a self-energy-like object for non-canonical degrees of freedom. We further make a resummation of density correlations to obtain analogues of the RPA and GW approximations commonly employed for canonical degrees of freedom. We discuss difficulties related to generating higher-order approximations which are consistent with conservation laws, which represents an outstanding issue. We also discuss how the interplay between canonical and non-canonical degrees of freedom offers a useful paradigm for organising the phase diagram of correlated electronic behaviour.

# 1 Introduction

The task of understanding interacting quantum systems is an inherently challenging one, as the complexity of a quantum system increases exponentially with size. Nevertheless our microscopic understanding of the world is built upon quantum foundations. A cornerstone of this success is the semi-classical notion of a quasi-particle, which reflects the organisation of correlations around underlying quantum degrees of freedom (DOF). As with all modelling, the identification of appropriate DOFs permits the most relevant correlations to be isolated, allowing for an accurate and efficient description of a system.

A quantum DOF is specified by the algebra it obeys. Correlations in a system are induced through the action of the Hamiltonian, as dictated by Heisenberg's equation of motion $\frac{\mathrm{d}}{\mathrm{d}t}\mathcal{O} = \mathrm{i}[\boldsymbol{H}, \mathcal{O}]$, and it is the DOF's algebra which governs how correlations are organised (here and throughout the text we adopt units such that $\hbar = 1$).

The best understood quantum DOFs are the canonical ones, bosons and fermions. These obey the familiar algebra of the form

$$[\boldsymbol{a}, \boldsymbol{a}^\dagger]_\mp = 1, \quad [\boldsymbol{n}, \boldsymbol{a}^\dagger] = \boldsymbol{a}^\dagger, \quad [\boldsymbol{n}, \boldsymbol{a}] = -\boldsymbol{a}, \tag{1}$$

with $[\boldsymbol{a}, \boldsymbol{a}^\dagger]_\mp = \boldsymbol{a}\boldsymbol{a}^\dagger \mp \boldsymbol{a}^\dagger\boldsymbol{a}$ and $[\cdot, \cdot] = [\cdot, \cdot]_-$. The defining characteristic of the canonical algebra is that the first relation yields the trivial operator. This has the consequence that a non-interacting Hamiltonian, i.e. $\boldsymbol{H}_0 = \sum_{i,j} \varepsilon_{ij} \boldsymbol{a}_i^\dagger \boldsymbol{a}_j$, has a linear action on the $\boldsymbol{a}_i^\dagger$, i.e. $[\boldsymbol{H}_0, \boldsymbol{a}_i^\dagger] = \sum_j \varepsilon_{ij} \boldsymbol{a}_j^\dagger$, which allows for the straightforward identification of single-particle modes. Interactions by contrast induce non-linear terms, which serve to impede a single-particle description. The semi-classical ideology is encapsulated by the Dyson equation

$$G = G_0 + G_0 \Sigma G, \tag{2}$$

which relates the single-particle Green's function $G$ of an interacting system to that of a corresponding non-interacting system $G_0$, together with a self-energy $\Sigma$ which encodes the interactions. In principle for model systems one can obtain an exact closed expression for $\Sigma$ (see e.g. Eq. (5-25b) of Ref. [1]). In practice $\Sigma$ is computed in a perturbative manner, where the lowest order contributions, or certain resummations thereof, characterise the dressing of the single-particle modes as quasi-particles. Behaviour which is well described in this way is commonly referred to as 'weakly correlated'. In particular this provides the microscopic foundation of Landau's Fermi liquid theory, the scheme by which the electronic properties of a wide variety of systems are understood.

The purpose of this manuscript is to examine whether non-canonical algebras can also provide useful quantum DOFs. Specifically we consider a family of algebras of the form

$$[\boldsymbol{a}, \boldsymbol{a}^\dagger]_\mp = 1 - \lambda \boldsymbol{n}, \quad [\boldsymbol{n}, \boldsymbol{a}^\dagger] = \boldsymbol{a}^\dagger, \quad [\boldsymbol{n}, \boldsymbol{a}] = -\boldsymbol{a}, \tag{3}$$

where $\lambda$ is a scalar parameter which is inherent to the algebra. Such algebras appear naturally for spin, electronic and local moment systems. The non-canonical contribution $\lambda \boldsymbol{n}$ presents an

immediate obstruction to invoking a representative non-interacting system, as in this case the action $[\boldsymbol{H}_0, \boldsymbol{a}_i^\dagger] = \sum_j \varepsilon_{ij}(1 - \lambda \boldsymbol{n}_i)\boldsymbol{a}_j^\dagger$ is non-linear. This is not such a severe restriction however, as single-particle modes can nevertheless be naturally identified at a mean-field level, in a manner analogous to the Hartree/Hartree–Fock approximations commonly employed for canonical DOFs. The challenging question is whether it is possible to systematically organise the remaining correlations, e.g. as done through the self-energy in the canonical case, so as to obtain an effective quasi-particle formalism. There is as yet no definitive consensus, but substantial progress has been made.

The legitimacy of non-canonical DOFs is a question of central importance for condensed matter physics. It is well established that the weakly correlated paradigm is insufficient to capture the great wealth of behaviour of interacting quantum systems, and non-canonical DOFs provide one of the most promising routes for characterising a range of the most interesting phenomena. Potential candidates include the pseudogap regime of the cuprates [2,3], heavy-fermion behaviour [4], and also indeed magnetically ordered systems [5–10]. In addition, competition between canonical and non-canonical quasi-particle descriptions may account for the emergence of quantum criticality, as outlined in the Discussion.

## 1.1  Historical summary

Efforts to employ non-canonical algebras for organising correlations in interacting quantum systems have a long and rich history. They arise naturally for spin systems, as spins are inherently governed by the non-canonical algebra $su(2)$. They were introduced for electronic systems by Hubbard [11], who employed the graded algebra of local projection operators to obtain approximations for the electronic Green's function of the Hubbard model [11–13].

There are three prominent lines of approach:

- The first is based around the operator projection/equations of motion methods of Mori [14,15], Zwanzig [16], and Tserkovnikov [17], which organise correlations in hierarchies of orthogonality around mean-field single-particle modes. A key issue here is when and how to truncate the higher order contributions. Very early efforts along these lines are found in the works of Rowe [18] and Roth [19], while more recent efforts can be found in [20–26]. There are in particular two bodies of work which have extensively investigated strongly correlated electronic behaviour in this manner, that of Plakida and coworkers [2,27–30], and the composite operator method of Avella–Mancini [3,31]. These studies go a long way towards demonstrating the promise of non-canonical DOFs.

- A second line of approach is built upon organising correlations in an expansion about the atomic limit, or high-temperature limit. For a spin model this can be viewed equivalently as the high-field limit, and for the electronic lattice models as the limit of strong on-site repulsion (or attraction). Isolated sites admit exact mean-field solution, and a detailed diagrammatic framework has been developed for incorporating interactions between sites [7,8,32–36]. This technique has proved complicated to implement in practice however, which has limited its application in general.

- The third line of approach utilises the Schwinger method [1, 37], where correlations are related to fluctuations induced by the inclusion of external sources. Here correlations may be organised by arranging variations of the external sources order by order in a perturbative expansion. To this end non-canonical algebras of the form of Eq. (3) are particularly useful, as they come endowed with an internal parameter $\lambda$ which provides a natural means of organising the expansion. Efforts in this direction have advanced only relatively recently, see [38–43]. A challenge here is to identify which are in fact the best correlation functions to construct an expansion for, as non-canonical correlations obstruct the canonical expansion of the inverse Green's function. This is a direction that is far from having been broadly explored.

In this paper we align the first and third of these approaches. In doing so we derive an closed expression for a self-energy-like object for non-canonical DOFs, and furthermore obtain an approximation capturing the screening of density correlations.

## 1.2 Overview

The paper offers a detailed overview of the concept of non-canonical DOFs. We adopt a pedagogical style, highlighting connections between various perspectives. We also keep analogy with canonical DOFs as transparent as possible.

Firstly in Sec. 2 we summarise and discuss several important non-canonical algebras, providing examples for spin, electron, and local moment lattice models. We demonstrate in particular that the electronic DOF may be cast in either a canonical or non-canonical form. The first underlies the Landau quasi-particle description, while the latter corresponds to the approach initiated by Hubbard.

The core of the paper is comprised of Secs. 3–5. In Sec. 3 we introduce a simple representative model system which is useful for developing the non-canonical approach. We analyse this from the Mori–Zwanzig–Tserkovnikov perspective in Sec. 4, demonstrating that the single-particle Green's function of the non-canonical DOF admits a Dyson form, arranging correlations between mean-field single-particle modes and an irreducible self-energy-like object $M_p^\star(\omega)$. We then switch to the Schwinger method in Sec. 5. Building upon the analysis of Sec. 4, we derive a novel exact representation for $M_p^\star(\omega)$, taking a functional differential form. We discuss issues related to generating conserving approximations in Sec. 5.3, and consider routes by which this may be addressed. We make a resummation of density-induced correlations in Sec. 5.4, and obtain an approximation which can be viewed as an analogue of the RPA and GW approximations for a canonical DOF.

In Sec. 6 we provide a discussion where we explore the non-canonical paradigm on general grounds. We demonstrate that if one assumes that non-canonical algebras do indeed provide legitimate quantum DOFs, then one arrives at a powerful governing principle for the phase diagram of correlated electronic behaviour.

We conclude in Sec. 7. There follow two appendices: in App. A we provide a formal overview of the algebraic structures underlying the discussion of Sec. 2, and in App. B we highlight the connection between the analysis of Sec. 4 and the more familiar Mori–Zwanzig formalism.

# 2 Key examples of non-canonical degrees of freedom

In this section we discuss several important examples of non-canonical DOFs. These serve to offer both a foundation and a motivation for the subsequent sections. First we highlight the well-known fact that the spin may be regarded as a non-canonical DOF, see e.g. [5], despite that spin-wave theory is more commonly formulated through the Holstein–Primakoff/Dyson–Maleev mapping to canonical bosons. Then we consider the settings of electronic and local moment lattice models, and demonstrate in both cases that the electron may be formulated in two ways, either as a canonical DOF as underlies Landau's quasi-particle framework, or as a non-canonical DOF as advanced by Hubbard. Our discussions here combine and extend those of Refs. [4, 43]. To complement this section we provide a general overview of the underlying algebraic structures, and their representations, in App. A.

## 2.1 Spin

A spin provides the simplest example of a non-canonical DOF, and one we will consider throughout the paper.

First we remind that an isolated spin is governed by the non-canonical algebra $su(2)$,

$$[\boldsymbol{S}^\alpha, \boldsymbol{S}^\beta] = i\epsilon^{\alpha\beta\gamma}\boldsymbol{S}^\gamma, \tag{4}$$

with $\alpha, \beta, \gamma \in \{x, y, z\}$. It has a basis of of $2S + 1$ states, where $S$ is the magnitude of the spin fixed through the Casimir identity $\vec{\boldsymbol{S}} \cdot \vec{\boldsymbol{S}} = S(S + 1)$, and $S$ is quantised in units of half-integer. The $su(2)$ algebra is commonly re-expressed through raising and lowing operators $\boldsymbol{S}^\pm = \boldsymbol{S}^x \pm i\boldsymbol{S}^y$, yielding the relations $[\boldsymbol{S}^+, \boldsymbol{S}^-] = 2\boldsymbol{S}^z$ and $[\boldsymbol{S}^z, \boldsymbol{S}^\pm] = \pm\boldsymbol{S}^\pm$. It is then convenient to label the basis states of a given spin by the eigenvalues of $\boldsymbol{S}^z$, which take values $-S, -S + 1, -S + 2, \ldots, S$.

A spin can be interpreted as a non-canonical DOF in the sense of Eq. (3) through the identification

$$\boldsymbol{a} = \tfrac{1}{\sqrt{2S}}\boldsymbol{S}^+, \quad \boldsymbol{a}^\dagger = \tfrac{1}{\sqrt{2S}}\boldsymbol{S}^-, \quad \boldsymbol{n} = S - \boldsymbol{S}^z, \tag{5}$$

after which the algebraic relations Eq. (4) become

$$[\boldsymbol{a}, \boldsymbol{a}^\dagger] = 1 - \tfrac{1}{S}\boldsymbol{n}, \quad [\boldsymbol{n}, \boldsymbol{a}^\dagger] = \boldsymbol{a}^\dagger, \quad [\boldsymbol{n}, \boldsymbol{a}] = -\boldsymbol{a}. \tag{6}$$

Here $1/S$ plays the role of $\lambda$, offering a parameter for organising correlations. The operator $\boldsymbol{n}$ has eigenvalues $0, 1, 2, \ldots, 2S$, with $0$ corresponding to a spin polarised in the $z$-direction, and low values of $\langle \boldsymbol{n} \rangle$ corresponding to small deviations from this. We can thus anticipate that this non-canonical DOF offers a quasi-particle description appropriate for magnetically ordered systems, i.e. spin-wave theory.

An expression for $\boldsymbol{n}$ is obtained as a rewriting of the Casimir identity

$$\boldsymbol{n} = \boldsymbol{a}^\dagger\boldsymbol{a} + \tfrac{1}{2S}\boldsymbol{n}(\boldsymbol{n} - 1). \tag{7}$$

For simplest case of $S = 1/2$ this reduces to $\boldsymbol{n} = \boldsymbol{a}^\dagger\boldsymbol{a}$.

We remark however that spin-wave theory is *not* conventionally formulated upon Eq. (6). In the absence of a consensus on the legitimacy of non-canonical DOFs, the spin is instead most commonly treated within the canonical quasi-particle framework as in the approaches of Holstein–Primakoff [44] and Dyson–Maleev [45, 46]. This is achieved by employing a representation of the spin generators in terms of canonical bosons $\boldsymbol{b}$ as follows

$$\boldsymbol{S}^+ = \sqrt{2S}\big(1 - \tfrac{1}{2S}\boldsymbol{b}^\dagger\boldsymbol{b}\big)^{1-\gamma}\boldsymbol{b}, \quad \boldsymbol{S}^- = \sqrt{2S}\,\boldsymbol{b}^\dagger\Big(1 - \tfrac{1}{2S}\boldsymbol{b}^\dagger\boldsymbol{b}\Big)^{\gamma}, \quad \boldsymbol{S}^z = S - \boldsymbol{b}^\dagger\boldsymbol{b}, \tag{8}$$

with $\gamma = 1/2$ for Holstein–Primakoff and $\gamma = 0$ for Dyson–Maleev. Again correlations are organised around the large-$S$ limit. While this approach has proved highly effective in practice, it lacks the robustness of a true weakly correlated theory. The kinetic term still induces correlations, and these obstruct application of the canonical diagrammatic framework. Further issues stem from the fact that the representative bosonic system includes multitudes of unphysical states, as a single boson has infinitely many basis states while a spin has a strictly finite number. Although the algebraic relations of the operators in Eq. (8) close on the physical Hilbert space, there is no clear way to maintain this restriction when analysing dynamical correlations at finite-temperature based on a $1/S$ expansion. By contrast, there are no unphysical states for the above non-canonical formulation of the spin DOF. The challenge instead is how to account for the non-canonical contribution in Eq. (6), which is the core focus of this paper.

## 2.2   Electron

We now turn our attention to the setting of electronic lattice models. An isolated electronic site has a basis of four states

$$|\circ\rangle, \quad |\!\downarrow\rangle, \quad |\!\uparrow\rangle, \quad |\bullet\rangle, \tag{9}$$

corresponding respectively to an empty site, a singly occupied site with spin either down or up, and a doubly occupied site. A key point we wish to stress is that there are two distinct ways of interpreting the electronic DOF, one of which takes the canonical form of Eq. (1), and another which takes the non-canonical form of Eq. (3).

The canonical formulation is simple and familiar. The electronic DOF is expressed through the algebra

$$\{\boldsymbol{c}_\sigma, \boldsymbol{c}_{\sigma'}^\dagger\} = \delta_{\sigma\sigma'}, \quad [\boldsymbol{n}_\sigma, \boldsymbol{c}_{\sigma'}^\dagger] = \delta_{\sigma\sigma'}\boldsymbol{c}_\sigma^\dagger, \quad [\boldsymbol{n}_\sigma, \boldsymbol{c}_{\sigma'}] = -\delta_{\sigma\sigma'}\boldsymbol{c}_\sigma, \tag{10}$$

with $\boldsymbol{n}_\sigma = \boldsymbol{c}_\sigma^\dagger\boldsymbol{c}_\sigma$ and $\sigma \in \{\downarrow, \uparrow\}$. We can view this as grouping the four electronic states as two copies of canonical fermions

$$\big\{\,|0\rangle\,; |\!\downarrow\rangle\,\big\} \otimes \big\{\,|0\rangle\,; |\!\uparrow\rangle\,\big\}, \tag{11}$$

where the semicolon denotes a relative grading of the states to either side. This canonical way of characterising the electronic DOF offers one route to organise electronic correlations in interacting systems. In particular it underlies our understanding of conventional metals as Landau Fermi liquids.

The non-canonical formulation is less simple and less familiar. This algebra mixes all four

electronic basis states, and to describe it let us group the states as follows

$$\{ \left| \downarrow \right\rangle , \left| \uparrow \right\rangle ; \ \left| \circ \right\rangle , \left| \bullet \right\rangle \}. \tag{12}$$

The algebra is composed first of a set of $su(2)$ generators $\vec{s}$ acting on the spin-doublet $\{\left| \downarrow \right\rangle , \left| \uparrow \right\rangle\}$, a second set of $su(2)$ generators $\vec{\eta}$ acting on the charge-doublet $\{\left| \circ \right\rangle , \left| \bullet \right\rangle\}$, and a generator $\boldsymbol{\theta}$ which weights the two doublets oppositely, and thus commutes with both $\vec{s}$ and $\vec{\eta}$. In addition there are 8 fermionic generators $\boldsymbol{q}_{\sigma\nu}$ and $\boldsymbol{q}_{\sigma\nu}^{\dagger}$, with $\sigma \in \{\downarrow, \uparrow\}$ and $\nu \in \{\circ, \bullet\}$, which act between the two doublets. The non-zero anti-commutation relations of the $\boldsymbol{q}_{\sigma\nu}$ are

$$\begin{aligned}
\{\boldsymbol{q}_{\sigma\nu}, \boldsymbol{q}_{\sigma\nu}^{\dagger}\} &= \tfrac{1+\kappa^2}{4} - \kappa\,(\sigma\boldsymbol{s}^z - \nu\boldsymbol{\eta}^z), \\
\{\boldsymbol{q}_{\downarrow\nu}, \boldsymbol{q}_{\uparrow\nu}^{\dagger}\} &= \kappa\,\boldsymbol{s}^+, \qquad \{\boldsymbol{q}_{\sigma\circ}, \boldsymbol{q}_{\sigma\bullet}^{\dagger}\} = -\kappa\,\boldsymbol{\eta}^+, \\
\{\boldsymbol{q}_{\uparrow\nu}, \boldsymbol{q}_{\downarrow\nu}^{\dagger}\} &= \kappa\,\boldsymbol{s}^-, \qquad \{\boldsymbol{q}_{\sigma\bullet}, \boldsymbol{q}_{\sigma\circ}^{\dagger}\} = -\kappa\,\boldsymbol{\eta}^-, \\
\{\boldsymbol{q}_{\sigma\nu}, \boldsymbol{q}_{\sigma'\nu'}\} &= \{\boldsymbol{q}_{\sigma\nu}^{\dagger}, \boldsymbol{q}_{\sigma'\nu'}^{\dagger}\} = \tfrac{1-\kappa^2}{4}\epsilon_{\sigma\sigma'}\epsilon_{\nu\nu'},
\end{aligned} \tag{13}$$

where $\sigma$ takes values $-1, 1$ for $\sigma = \downarrow, \uparrow$, and $\nu$ takes values $-1, 1$ for $\nu = \circ, \bullet$, and $\epsilon_{\downarrow\uparrow} = -\epsilon_{\uparrow\downarrow} = \epsilon_{\circ\bullet} = -\epsilon_{\bullet\circ} = 1$. The commutation relations of the $\boldsymbol{q}_{\sigma\nu}$ with $\vec{s}$ are

$$\begin{aligned}
[\boldsymbol{s}^z, \boldsymbol{q}_{\sigma\nu}^{\dagger}] &= \frac{\sigma}{2}\boldsymbol{q}_{\sigma\nu}^{\dagger}, \qquad [\boldsymbol{s}^+, \boldsymbol{q}_{\downarrow\nu}^{\dagger}] = -\boldsymbol{q}_{\uparrow\nu}^{\dagger}, \qquad [\boldsymbol{s}^-, \boldsymbol{q}_{\uparrow\nu}^{\dagger}] = -\boldsymbol{q}_{\downarrow\nu}^{\dagger}, \\
[\boldsymbol{s}^z, \boldsymbol{q}_{\sigma\nu}] &= -\frac{\sigma}{2}\boldsymbol{q}_{\sigma\nu}, \qquad [\boldsymbol{s}^+, \boldsymbol{q}_{\uparrow\nu}] = \boldsymbol{q}_{\downarrow\nu}, \qquad [\boldsymbol{s}^-, \boldsymbol{q}_{\downarrow\nu}] = \boldsymbol{q}_{\uparrow\nu},
\end{aligned} \tag{14}$$

with $\vec{\eta}$ are

$$\begin{aligned}
[\boldsymbol{\eta}^z, \boldsymbol{q}_{\sigma\nu}^{\dagger}] &= \frac{\nu}{2}\boldsymbol{q}_{\sigma\nu}^{\dagger}, \qquad [\boldsymbol{\eta}^+, \boldsymbol{q}_{\sigma\circ}^{\dagger}] = -\boldsymbol{q}_{\sigma\bullet}^{\dagger}, \qquad [\boldsymbol{\eta}^-, \boldsymbol{q}_{\sigma\bullet}^{\dagger}] = -\boldsymbol{q}_{\sigma\circ}^{\dagger}, \\
[\boldsymbol{\eta}^z, \boldsymbol{q}_{\sigma\nu}] &= -\frac{\nu}{2}\boldsymbol{q}_{\sigma\nu}, \qquad [\boldsymbol{\eta}^+, \boldsymbol{q}_{\sigma\bullet}] = \boldsymbol{q}_{\sigma\circ}, \qquad [\boldsymbol{\eta}^-, \boldsymbol{q}_{\sigma\circ}] = \boldsymbol{q}_{\sigma\bullet},
\end{aligned} \tag{15}$$

and with $\boldsymbol{\theta}$ are

$$\begin{aligned}
[\boldsymbol{\theta}, \boldsymbol{q}_{\sigma\nu}^{\dagger}] &= \tfrac{1+\kappa^2}{4\kappa}\,\boldsymbol{q}_{\sigma\nu}^{\dagger} - \tfrac{1-\kappa^2}{4\kappa}\epsilon_{\sigma\sigma'}\epsilon_{\nu\nu'}\boldsymbol{q}_{\sigma'\nu'}, \\
[\boldsymbol{\theta}, \boldsymbol{q}_{\sigma\nu}] &= -\tfrac{1+\kappa^2}{4\kappa}\,\boldsymbol{q}_{\sigma\nu} + \tfrac{1-\kappa^2}{4\kappa}\epsilon_{\sigma\sigma'}\epsilon_{\nu\nu'}\boldsymbol{q}_{\sigma'\nu'}^{\dagger}.
\end{aligned} \tag{16}$$

We can regard these relations as a matrix version of the non-canonical form of Eq. (3), with the $\boldsymbol{q}_{\sigma\nu}$ playing the role of $\boldsymbol{a}$, and the generators $\vec{s}$, $\vec{\eta}$, $\boldsymbol{\theta}$ playing the role of $\boldsymbol{n}$. The continuous parameter $\kappa$ then plays the role of $\lambda$.

There are a number of remarks to be made. Firstly we emphasise that this non-canonical formulation of the electron has a natural origin. It may be viewed as a generalisation of $su(2)$ to the electronic setting, see App. A. Formally we may call the canonical form of Eq. (10) above as $u(1|1) \otimes u(1|1)$, and this non-canonical formulation as $u(2|2)$, where the vertical line indicates the graded structure of the algebra. Here the electron corresponds to the fundamental representation of $u(2|2)$, just as spin-1/2 corresponds to the fundamental representation of $su(2)$.

The continuous parameter $\kappa$ couples to the non-canonical terms in Eq. (13), allowing us to organise the correlations they induce. It is perhaps worthwhile to highlight that the appearance of

$\kappa$ is in fact highly non-trivial. It has an exceptional origin as discussed in App. A, which can be traced to the exceptional Lie superalgebra $d(2, 1; \epsilon)$, see e.g. App. A of Ref. [47]. The $\kappa$-dependence of the algebra underlies the integrability of the Hubbard model in 1D [48–50], and plays a similarly crucial role in instances of quantum integrability in the AdS/CFT correspondence [51, 52].

It may not be immediately obvious that the algebra comprised of Eqs. (13)-(16) closes on the four electronic basis states. This is seen however by expressing the generators in terms of spinful canonical fermions as follows

$$
\begin{aligned}
&\boldsymbol{q}_{\sigma\circ}^{\dagger} = \tfrac{1+\kappa}{2}\, \boldsymbol{c}_{-\sigma} - \kappa\, \boldsymbol{n}_\sigma \boldsymbol{c}_{-\sigma}, &\qquad &\boldsymbol{q}_{\sigma\bullet}^{\dagger} = \sigma\big(\tfrac{1-\kappa}{2}\, \boldsymbol{c}_\sigma^{\dagger} + \kappa\, \boldsymbol{n}_{-\sigma} \boldsymbol{c}_\sigma^{\dagger}\big), \\
&\boldsymbol{q}_{\sigma\circ} = \tfrac{1+\kappa}{2}\, \boldsymbol{c}_{-\sigma}^{\dagger} - \kappa\, \boldsymbol{n}_\sigma \boldsymbol{c}_{-\sigma}^{\dagger}, &\qquad &\boldsymbol{q}_{\sigma\bullet} = \sigma\big(\tfrac{1-\kappa}{2}\, \boldsymbol{c}_\sigma + \kappa\, \boldsymbol{n}_{-\sigma} \boldsymbol{c}_\sigma\big), \\
&\boldsymbol{s}^{+} = \boldsymbol{c}_\uparrow^{\dagger} \boldsymbol{c}_\downarrow, &\quad \boldsymbol{s}^{-} = \boldsymbol{c}_\downarrow^{\dagger} \boldsymbol{c}_\uparrow, &\quad \boldsymbol{s}^{z} = \tfrac{1}{2}\big(\boldsymbol{n}_\uparrow - \boldsymbol{n}_\downarrow\big), \\
&\boldsymbol{\eta}^{+} = \boldsymbol{c}_\downarrow^{\dagger} \boldsymbol{c}_\uparrow^{\dagger}, &\quad \boldsymbol{\eta}^{-} = \boldsymbol{c}_\uparrow \boldsymbol{c}_\downarrow, &\quad \boldsymbol{\eta}^{z} = \tfrac{1}{2}\big(\boldsymbol{n}_\uparrow + \boldsymbol{n}_\downarrow - 1\big), \\
&\boldsymbol{\theta} = (\boldsymbol{n}_\downarrow - \tfrac{1}{2})(\boldsymbol{n}_\uparrow - \tfrac{1}{2}),
\end{aligned}
\tag{17}
$$

from which the algebraic relations Eqs. (13)-(16) can be directly verified. (The signs of $\boldsymbol{q}_{\sigma\bullet}$ and $\boldsymbol{q}_{\sigma\bullet}^{\dagger}$ are flipped with respect to Refs. [4, 43], which gives the algebra a more symmetric form.)

Ultimately to characterise the electronic properties of a system we wish to compute correlation functions of the $\boldsymbol{c}$. It is a somewhat remarkable fact that these follow immediately from the correlation functions of the $\boldsymbol{q}$. The inversion of Eq. (17) takes a linear form

$$
\boldsymbol{c}_\downarrow^{\dagger} = \boldsymbol{q}_{\uparrow\circ} - \boldsymbol{q}_{\downarrow\bullet}^{\dagger}, \qquad \boldsymbol{c}_\uparrow^{\dagger} = \boldsymbol{q}_{\downarrow\circ} + \boldsymbol{q}_{\uparrow\bullet}^{\dagger},
\tag{18}
$$

and so the $\boldsymbol{q}$ may be thought of as splitting the electron (as opposed to a fractionalisation). As a consequence, correlation functions of the $\boldsymbol{c}$ are obtained directly as linear combinations of correlation functions of the $\boldsymbol{q}$. In this way, the non-canonical formulation offers a second distinct route by which to organise electronic correlations. An essential difference with the canonical formulation is that the $\boldsymbol{q}$ lead to a splitting in two of the electronic band, driven by an interaction built from $\boldsymbol{\theta}$ obeying Eq. (16), which allows for the emergence of a Mott gap [11, 12, 43].

The non-canonical formulation of the electron thus offers the possibility of a quasi-particle description of an unconventional metal, i.e. one which is not a Landau Fermi liquid. Given the seminal contributions of Hubbard in this direction [11–13], it would be appropriate to term this a 'Hubbard Fermi liquid'. In general it is not possible to tell a priori whether a given system can be characterised as either a Landau or Hubbard Fermi liquid, in principle one should examine all known possible ways of organising correlations and identify which describe behaviour consistent with observations/simulations of the system.

Let us then consider conditions under which we may anticipate that the $\boldsymbol{q}$ do provide an effective quasi-particle description. Firstly, based on the appearance of the non-canonical terms in Eq. (13) we may expect the quasi-particle description is most robust when $\langle \vec{\boldsymbol{s}} \rangle \sim 0$ and $\langle \vec{\boldsymbol{\eta}} \rangle \sim 0$, i.e. for a paramagnetic state in the vicinity of half-filling[1]. Secondly, we note that a kinetic contribution to

---

[1]Indeed, that the splitting of the electron is a finite-density effect explains why it escapes the classification of elementary particles coming from high-energy physics.

the Hamiltonian density of the form

$$H_{ij}^{\text{kin}} = -\sum_{\sigma=\downarrow,\uparrow} \sum_{\nu=\circ,\bullet} t_\nu \left( \boldsymbol{q}_{i\sigma\nu}^\dagger \boldsymbol{q}_{j\sigma\nu} + \boldsymbol{q}_{j\sigma\nu}^\dagger \boldsymbol{q}_{i\sigma\nu} \right), \tag{19}$$

generically incorporates correlated hopping interactions, as discussed in detail in Ref. [43]. By contrast, correlations in hopping conflict with the canonical quasi-particle framework, obstructing the use of diagrammatic techniques. We may thus anticipate that their presence favours the non-canonical regime. Thirdly, we highlight that the Hubbard interaction can be expressed as $U \sum_i \boldsymbol{\theta}_i$, up to a shift of the chemical potential. As the action of $\boldsymbol{\theta}$ in Eq. (16) is linear, we see that $U$ then plays a role akin to an additional chemical potential. That is, it does not induce correlations from the perspective of the $\boldsymbol{q}$. Instead it controls the splitting in two of the electronic band, differentiating the split electrons of Eq. (18). We see here no restriction on the value of $U$, but due to the singular nature of Eq. (16) we should take $U \to 0$ when considering $\kappa \to 0$. Finally, in systems with strong spin-orbit the spin and charge of the electron get coupled, and it would be interesting to investigate to what extent this drives correlations governed by the non-canonical formulation of the electron.

We next address a criticism sometimes made of the non-canonical formulation, which is that it generically leads to a violation of the Luttinger sum rule [53], see e.g. Refs. [13, 43]. The sum rule states that the volume enclosed by the Fermi surface is directly proportional to the electron density, and independent of interactions. Luttinger's proof however is tied to the Luttinger–Ward functional, which is defined through the canonical perturbative expansion [54]. A later non-perturbative proof due to Oshikawa is also often cited in this context [55], but this proof also makes the crucial assumption that the underlying DOF is canonical (see e.g. Eq. (5) of [55]). The non-canonical formulation of the electronic DOF lies outside the realm of these proofs. Here as $U$ acts as an *additional* chemical potential for the $\boldsymbol{q}$, via Eq. (16), the non-canonical formulation displays a broader class of mean-field single-particle modes than the canonical formulation, and the Luttinger sum rule is generically violated for $U \neq 0$. There is no inconsistency. Instead, violation of the Luttinger sum rule can be regarded as an observable feature which distinguishes the Hubbard Fermi liquid from the more conventional Landau Fermi liquid. Indeed this violation inidcates that the Hubbard Fermi liquid is not adiabatically connected to the Landau Fermi liquid, a topic which is further explored in the Discussion.

To connect to earlier literature we highlight that the non-canonical relations above, Eqs. (13)-(16), are equivalent to the Hubbard algebra when $\kappa = 1$. That is, we can introduce the Hubbard operators [11] through

$$\begin{aligned}
\boldsymbol{X}_{\nu\sigma} &= \bar{\nu}\boldsymbol{q}_{\bar{\sigma}\nu}^\dagger, \quad \boldsymbol{X}_{\nu\nu} = \nu\boldsymbol{\eta}^z + \boldsymbol{\theta} + 1/4, \quad \boldsymbol{X}_{\bullet\circ} = \boldsymbol{\eta}^+, \quad \boldsymbol{X}_{\circ\bullet} = \boldsymbol{\eta}^-, \\
\boldsymbol{X}_{\sigma\nu} &= \bar{\nu}\boldsymbol{q}_{\bar{\sigma}\nu}, \quad \boldsymbol{X}_{\sigma\sigma} = \sigma\boldsymbol{s}^z - \boldsymbol{\theta} + 1/4, \quad \boldsymbol{X}_{\uparrow\downarrow} = \boldsymbol{s}^+, \quad \boldsymbol{X}_{\downarrow\uparrow} = \boldsymbol{s}^-,
\end{aligned} \tag{20}$$

evaluated at $\kappa = 1$, with $\bar{\sigma} = -\sigma$ and $\bar{\nu} = -\nu$. These satisfy the Hubbard algebra

$$[\boldsymbol{X}_{ab}, \boldsymbol{X}_{cd}]_\mp = \delta_{bc}\boldsymbol{X}_{ad} \mp \delta_{ad}\boldsymbol{X}_{cb}, \tag{21}$$

where the plus sign is only used when both operators in the bracket are fermionic.

Finally we remark upon a reduced description of electronic systems frequently employed, for example in the context of the $t$-$J$ model, which projects out the doubly occupied site $|\bullet\rangle$ from the electronic basis of states. Here one may consider employing the sub-algebra generated by $\boldsymbol{q}_{\sigma\circ}$ and $\boldsymbol{q}_{\sigma\circ}^{\dagger}$. We highlight however this admits a 3-dimensional representation only when $\kappa = \pm1$, while more generally the smallest non-trivial representation corresponds to the 4-dimensional electronic basis. The parameter $\kappa$ is thus not available to organise correlations on the reduced Hilbert space in a manner independent of the full electronic formulation. Indeed we see from Eq. (18) that we need incorporate $\boldsymbol{q}_{\sigma\bullet}$ and $\boldsymbol{q}_{\sigma\bullet}^{\dagger}$ in order to access the true electronic correlations.

## 2.3  Local moment

The third and final setting we consider are local moment systems. Specifically we mean effective lattice models which have at each site both an electron and a spin (referred to as a spin-moment to distinguish it from the electronic spin). Here an isolated site has a basis of $4 \times (2S + 1)$ states, where $S$ is the magnitude of the spin-moment. As in the previous electronic case, there are again two distinct ways of expressing the local DOF, which cast the electron through either a canonical or non-canonical algebra.

The canonical formulation is again the conventional one. Here the electron and the spin are treated as independent. That is, with the electrons governed by $\boldsymbol{c}_{\sigma}$ obeying the canonical relations of Eq. (10), and the spin-moment governed by $\vec{\boldsymbol{S}}$ obeying the $su(2)$ relations of Eq. (4). Formally we can denote the combined DOF as $u(1|1) \otimes u(1|1) \otimes su(2)$, i.e. as two species of fermion and a spin. One may expect that this characterisation is appropriate for behaviour where the electrons form a Landau Fermi liquid and the spin-moments are free to order.

The non-canonical formulation of the local moment DOF can be understood as a higher dimensional representation of the non-canonical electronic DOF. Formally it is again the algebra $u(2|2)$, which now mixes all $4 \times (2S + 1)$ basis states. Here the spin-moment $\vec{\boldsymbol{S}}$ gets entwined with the electronic spin $\vec{\boldsymbol{s}}$, and it is the total spin operator

$$\vec{\boldsymbol{\Sigma}} = \vec{\boldsymbol{s}} + \vec{\boldsymbol{S}}, \tag{22}$$

which enters the algebra. In addition to $\vec{\boldsymbol{\Sigma}}$ there are again $\vec{\boldsymbol{\eta}}$ and $\boldsymbol{\theta}$, along with fermionic generators $\boldsymbol{q}_{\sigma\nu}$ and $\boldsymbol{q}_{\sigma\nu}^{\dagger}$. The non-zero anti-commutation relations of $\boldsymbol{q}_{\sigma\nu}$ are here

$$\begin{aligned}
\{\boldsymbol{q}_{\sigma\nu}, \boldsymbol{q}_{\sigma\nu}^{\dagger}\} &= \tfrac{1+\kappa^2}{4} - \tfrac{\kappa}{2S+1}(\sigma\boldsymbol{\Sigma}^z - \nu\boldsymbol{\eta}^z), \\
\{\boldsymbol{q}_{\downarrow\nu}, \boldsymbol{q}_{\uparrow\nu}^{\dagger}\} &= \tfrac{\kappa}{2S+1}\boldsymbol{\Sigma}^+, & \{\boldsymbol{q}_{\sigma\circ}, \boldsymbol{q}_{\sigma\bullet}^{\dagger}\} &= -\tfrac{\kappa}{2S+1}\boldsymbol{\eta}^+, \\
\{\boldsymbol{q}_{\uparrow\nu}, \boldsymbol{q}_{\downarrow\nu}^{\dagger}\} &= \tfrac{\kappa}{2S+1}\boldsymbol{\Sigma}^-, & \{\boldsymbol{q}_{\sigma\bullet}, \boldsymbol{q}_{\sigma\circ}^{\dagger}\} &= -\tfrac{\kappa}{2S+1}\boldsymbol{\eta}^-, \\
\{\boldsymbol{q}_{\sigma\nu}, \boldsymbol{q}_{\sigma'\nu'}\} &= \{\boldsymbol{q}_{\sigma\nu}^{\dagger}, \boldsymbol{q}_{\sigma'\nu'}^{\dagger}\} = \tfrac{1-\kappa^2}{4}\epsilon_{\sigma\sigma'}\epsilon_{\nu\nu'}.
\end{aligned} \tag{23}$$

The commutation relations of $\boldsymbol{q}_{\sigma\nu}$ with $\vec{\boldsymbol{\Sigma}}$ are identical to those of Eq. (14) with $\vec{\boldsymbol{\Sigma}}$ replacing $\vec{\boldsymbol{s}}$, and the commutation relations of $\boldsymbol{q}_{\sigma\nu}$ with $\vec{\boldsymbol{\eta}}$ and $\boldsymbol{\theta}$ are given by Eqs. (15) and (16) respectively. Thus again we obtain a matrix version of the non-canonical form of Eq. (3). The non-canonical electronic DOF above is included here as the case $S = 0$, for which $\vec{\boldsymbol{\Sigma}} = \vec{\boldsymbol{s}}$.

In Eq. (23) we have two parameters which play the role of $\lambda$ in Eq. (3), both the continuous parameter $\kappa$ and the discrete parameter $\frac{1}{2S+1}$. We may again anticipate that employing $\kappa$ to organise correlations offers a quasi-particle description of an unconventional metal, i.e. an analogue/another instance of the 'Hubbard Fermi liquid'. As argued in Ref. [4], this non-canonical formulation may be appropriate for characterising heavy-fermion behaviour found in local moment systems. On the other hand, we may expect that employing $\frac{1}{2S+1}$ to organise correlations offers a quasi-particle description for magnetically ordered behaviour, useful for both electronic and local moment systems. We further remark that as opposed to entwining a spin-moment with the electronic spin, one could instead consider entwining a charge-moment with the electronic charge. This would offer a distinct large-$S$ limit, providing a means of characterising charge order driven by electronic correlations. Neither of these two routes to treating electronic ordering have been explored in any detail.

The non-canonical $\boldsymbol{q}$ of Eq. (23) again manifest a splitting of the electron

$$\boldsymbol{c}_\downarrow^\dagger = \boldsymbol{q}_{\uparrow\circ} - \boldsymbol{q}_{\downarrow\bullet}^\dagger, \qquad \boldsymbol{c}_\uparrow^\dagger = \boldsymbol{q}_{\downarrow\circ} + \boldsymbol{q}_{\uparrow\bullet}^\dagger. \tag{24}$$

Here the $\boldsymbol{q}$ are given explicitly by

$$\begin{aligned}
\boldsymbol{q}_{\downarrow\circ}^\dagger &= \tfrac{1}{2}\boldsymbol{c}_\uparrow + \tfrac{\kappa}{2S+1}\big(\tfrac{1}{2}\boldsymbol{c}_\uparrow - \boldsymbol{n}_\downarrow \boldsymbol{c}_\uparrow + \boldsymbol{c}_\downarrow \boldsymbol{S}^- + \boldsymbol{c}_\uparrow \boldsymbol{S}^z\big), \\
\boldsymbol{q}_{\uparrow\circ}^\dagger &= \tfrac{1}{2}\boldsymbol{c}_\downarrow + \tfrac{\kappa}{2S+1}\big(\tfrac{1}{2}\boldsymbol{c}_\downarrow - \boldsymbol{n}_\uparrow \boldsymbol{c}_\downarrow + \boldsymbol{c}_\uparrow \boldsymbol{S}^+ - \boldsymbol{c}_\downarrow \boldsymbol{S}^z\big), \\
\boldsymbol{q}_{\downarrow\bullet}^\dagger &= -\tfrac{1}{2}\boldsymbol{c}_\downarrow^\dagger + \tfrac{\kappa}{2S+1}\big(\tfrac{1}{2}\boldsymbol{c}_\downarrow^\dagger - \boldsymbol{n}_\uparrow \boldsymbol{c}_\downarrow^\dagger + \boldsymbol{c}_\uparrow^\dagger \boldsymbol{S}^- - \boldsymbol{c}_\downarrow^\dagger \boldsymbol{S}^z\big), \\
\boldsymbol{q}_{\uparrow\bullet}^\dagger &= \tfrac{1}{2}\boldsymbol{c}_\uparrow^\dagger - \tfrac{\kappa}{2S+1}\big(\tfrac{1}{2}\boldsymbol{c}_\uparrow^\dagger - \boldsymbol{n}_\downarrow \boldsymbol{c}_\uparrow^\dagger + \boldsymbol{c}_\downarrow^\dagger \boldsymbol{S}^+ + \boldsymbol{c}_\uparrow^\dagger \boldsymbol{S}^z\big),
\end{aligned} \tag{25}$$

along with their hermitian conjugates. To verify Eq. (23) algebraically it is necessary to employ the Casimir identity $\vec{\boldsymbol{S}} \cdot \vec{\boldsymbol{S}} = S(S+1)$. Both $\vec{\boldsymbol{s}}$ and $\vec{\boldsymbol{\eta}}$ take the same form as in the electronic case, Eq. (17), while

$$\boldsymbol{\theta} = \tfrac{1}{2} - \tfrac{1}{2S+1}\Big(\vec{\boldsymbol{\Sigma}} \cdot \vec{\boldsymbol{\Sigma}} + \tfrac{1}{3}\vec{\boldsymbol{\eta}} \cdot \vec{\boldsymbol{\eta}}\Big), \tag{26}$$

also gets modified. We may rewrite this final equation (employing $\vec{\boldsymbol{S}} \cdot \vec{\boldsymbol{S}} = S(S+1)$ and $\vec{\boldsymbol{s}} \cdot \vec{\boldsymbol{s}} + \vec{\boldsymbol{\eta}} \cdot \vec{\boldsymbol{\eta}} = \tfrac{3}{4}$) to express the Kondo coupling operator through $\boldsymbol{\theta}$ as follows

$$\vec{\boldsymbol{s}} \cdot \vec{\boldsymbol{S}} = \tfrac{1}{3}\vec{\boldsymbol{\eta}} \cdot \vec{\boldsymbol{\eta}} - \tfrac{2S+1}{2}\boldsymbol{\theta} - \tfrac{1+4S^2}{8}. \tag{27}$$

which clarifies how it acts on the $\boldsymbol{q}$.

## 3  A model system

Having outlined three important examples of non-canonical DOFs in the previous section, our goal now is to understand how they can be employed to organise correlations.

To proceed it is convenient to introduce a representative model system, so as to keep both notations and discussion as simple as need be. We thus consider a model on a $d$-dimensional

hypercubic lattice with Hamiltonian

$$\boldsymbol{H} = \sum_{i,j} \varepsilon_{ij} \boldsymbol{a}_i^\dagger \boldsymbol{a}_j + \tfrac{1}{2} \sum_{i,j} V_{ij} \boldsymbol{n}_i \boldsymbol{n}_j - \mu \sum_i \boldsymbol{n}_i, \tag{28}$$

expressed through a non-canonical DOF obeying

$$[\boldsymbol{a}_i, \boldsymbol{a}_j^\dagger]_\mp = \delta_{ij}(1 - \lambda \boldsymbol{n}_i), \quad [\boldsymbol{n}_i, \boldsymbol{a}_j^\dagger] = \delta_{ij} \boldsymbol{a}_i^\dagger, \quad [\boldsymbol{n}_i, \boldsymbol{a}_j] = -\delta_{ij} \boldsymbol{a}_i, \tag{29}$$

and set $\boldsymbol{n}_i = \boldsymbol{a}_i^\dagger \boldsymbol{a}_i$. We consider the cases of bosonic and fermionic DOFs in parallel, and take the convention that in instances of a sign ambiguity the upper/lower sign corresponds to bosonic/fermionic $\boldsymbol{a}_i$, $\boldsymbol{a}_i^\dagger$. We let $\mu$ control the on-site term, and so $\varepsilon_{ii} = V_{ii} = 0$. We assume full translational invariance, and for simplicity we take real $\varepsilon_{ij} = \varepsilon_{ji}$ and $V_{ij} = V_{ji}$, and will use these properties freely in the following.

Let us highlight that this is not an artificial system, but is in fact an important example. The above is a rewriting of the Heisenberg model of magnetism. Consider the Heisenberg Hamiltonian

$$\boldsymbol{H} = -\tfrac{J}{S} \sum_{\langle i,j \rangle} \vec{\boldsymbol{S}}_i \cdot \vec{\boldsymbol{S}}_j - h \sum_i \boldsymbol{S}_i^z, \tag{30}$$

where $\langle \cdot, \cdot \rangle$ denotes summation over nearest-neighbour pairs of sites. Employing Eq. (5) to re-express the spin as the non-canonical DOF of Eq. (6), for which $\lambda = 1/S$, this becomes (up to an additive constant)

$$\boldsymbol{H} = -J \sum_{\langle i,j \rangle} \left( \boldsymbol{a}_i^\dagger \boldsymbol{a}_j + \boldsymbol{a}_j^\dagger \boldsymbol{a}_i \right) - \tfrac{J}{S} \sum_{\langle i,j \rangle} \boldsymbol{n}_i \boldsymbol{n}_j + (h + zJ) \sum_i \boldsymbol{n}_i, \tag{31}$$

where $z = 2d$ is the coordination number of the lattice. As we take $\boldsymbol{n} = \boldsymbol{a}^\dagger \boldsymbol{a}$, we are focusing on the simplest $S = 1/2$ case. We can thus regard our model system as the spin-1/2 Heisenberg model, and our subsequent discussion can be read as an effort to formulate a non-canonical version of spin-wave theory. The description here corresponds to ferromagnetism for $J > 0$, and more complex ordering patterns are straightforwardly attained by appropriately orientating the $\boldsymbol{n}_i$ within the corresponding unit cell. Moreover, the model system also covers very general electronic and local moment models upon addition of a matrix notation, see e.g. [4, 43], but we do not incorporate this here in favour of simplicity. We emphasise though that in the electronic setting in particular we have a matrix form of $\boldsymbol{n} = \boldsymbol{a}^\dagger \boldsymbol{a}$ for all $\kappa$, and so a satisfactory quasi-particle description of our model system will yield a satisfactory quasi-particle description of an unconventional metal, i.e. the Hubbard Fermi liquid of Sec. 2.2.

We also highlight that our model system, and our subsequent analysis, reduces to the canonical case upon setting $\lambda = 0$.

In developing a quasi-particle description the central object of interest is the retarded single-particle Green's function

$$G_{ij}(t - t') = -i\vartheta(t - t') \langle [\boldsymbol{a}_i(t), \boldsymbol{a}_j^\dagger(t')] \rangle, \tag{32}$$

considered here at finite temperature $\langle \mathcal{O} \rangle = \frac{\text{Tr}(e^{-\beta \boldsymbol{H}} \mathcal{O})}{\text{Tr}(e^{-\beta \boldsymbol{H}})}$, with $\beta = 1/T$, and $\vartheta$ is the Heaviside function. In the next two sections we will obtain the same Dyson form for the Green's function along the lines of the first and third approaches outlined in Sec. 1.1 of the Introduction.

## 4  Structure of the retarded Green's function

Our objective is to understand how correlations may be organised around non-canonical DOFs. In this section we review how this is achieved along the lines of the Mori–Zwanzig–Tserkovnikov approach, which leads to an instructive Dyson form of the Green's function. Specifically, here we follow the formalism of Tserkovnikov [17] which results in the most transparent expressions. For completeness we highlight how this is related to the more familiar Mori–Zwanzig variant of the approach in App. B. We focus our discussion on the retarded Green's function, which is the physical correlation function we are ultimately interested in.

It is convenient to first introduce some compact notations [56]. We define two inner products on the space of operators

$$
\begin{aligned}
\langle \mathcal{O} | \mathcal{O}' \rangle &= \langle [\mathcal{O}, \mathcal{O}'] \rangle , \\
\langle\langle \mathcal{O}(t) | \mathcal{O}'(t') \rangle\rangle &= -\mathrm{i}\vartheta(t - t') \langle \mathcal{O}(t) | \mathcal{O}'(t') \rangle ,
\end{aligned}
\tag{33}
$$

in terms of which the retarded Green's function takes the form $G_{ij}(t - t') = \langle\langle \boldsymbol{a}_i(t) | \boldsymbol{a}_j^\dagger(t') \rangle\rangle$. Despite translational invariance in both time and space, it proves useful to keep both site indices and times explicit. We Fourier transform according to

$$
G_p(\omega) = \frac{1}{\mathcal{V}} \sum_{i,j} \int_0^\infty \mathrm{d}(t - t') \, e^{\mathrm{i}\omega t - \mathrm{i}p(i-j)} G_{ij}(t - t'), \qquad \text{Im}\,\omega > 0,
\tag{34}
$$

with $\mathcal{V}$ the total number of sites, or more generally as

$$
\begin{aligned}
\langle \mathcal{O}_p | \mathcal{O}'_p \rangle &= \frac{1}{\mathcal{V}} \sum_{i,j} e^{-\mathrm{i}p(i-j)} \langle \mathcal{O}_i | \mathcal{O}'_j \rangle , \\
\langle\langle \mathcal{O}_p | \mathcal{O}'_p \rangle\rangle_\omega &= \frac{1}{\mathcal{V}} \sum_{i,j} \int_0^\infty \mathrm{d}(t - t') \, e^{\mathrm{i}\omega(t - t') - \mathrm{i}p(i-j)} \langle\langle \mathcal{O}_i(t) | \mathcal{O}'_j(t') \rangle\rangle , \qquad \text{Im}\,\omega > 0.
\end{aligned}
\tag{35}
$$

The Green's function is determined through its equation of motion. For this we need evaluate

$$
\begin{aligned}
[\boldsymbol{H}, \boldsymbol{a}_i] &= \mu \boldsymbol{a}_i - \sum_l \varepsilon_{il}(1 - \lambda \boldsymbol{n}_i)\boldsymbol{a}_l - \sum_l V_{il} \boldsymbol{n}_l \boldsymbol{a}_i , \\
[\boldsymbol{H}, \boldsymbol{a}_i^\dagger] &= -\mu \boldsymbol{a}_i^\dagger + \sum_l \varepsilon_{li} \boldsymbol{a}_l^\dagger (1 - \lambda \boldsymbol{n}_i) + \sum_l V_{li} \boldsymbol{a}_i^\dagger \boldsymbol{n}_l ,
\end{aligned}
\tag{36}
$$

and it is useful to express these compactly as

$$[\boldsymbol{H}, \boldsymbol{a}_i] = -\sum_l (\tilde{\varepsilon}_{il} - \mu\delta_{il})\boldsymbol{a}_l - \boldsymbol{b}_i,$$

$$[\boldsymbol{H}, \boldsymbol{a}_i^\dagger] = \sum_l (\tilde{\varepsilon}_{li} - \mu\delta_{li})\boldsymbol{a}_l^\dagger + \boldsymbol{b}_i^\dagger, \tag{37}$$

where $\tilde{\varepsilon}_{ij}$ denotes an effective dispersion, and $\boldsymbol{b}_i$ denotes the remaining terms and in particular the non-linear contributions responsible for inducing correlations. Let us not make an attempt to specify $\tilde{\varepsilon}_{ij}$ for now.

We begin by taking the left equation of motion for $G_{ij}(t - t')$,

$$\begin{aligned}
\mathrm{i}\partial_t \langle\!\langle \boldsymbol{a}_i(t)|\boldsymbol{a}_j^\dagger(t')\rangle\!\rangle &= \delta(t - t') \langle \boldsymbol{a}_i|\boldsymbol{a}_j^\dagger\rangle + \langle\!\langle \mathrm{i}\dot{\boldsymbol{a}}_i(t)|\boldsymbol{a}_j^\dagger(t')\rangle\!\rangle \\
&= \delta(t - t') \langle \boldsymbol{a}_i|\boldsymbol{a}_j^\dagger\rangle + \sum_l (\tilde{\varepsilon}_{il} - \mu\delta_{il}) \langle\!\langle \boldsymbol{a}_l(t)|\boldsymbol{a}_j^\dagger(t')\rangle\!\rangle + \langle\!\langle \boldsymbol{b}_i(t)|\boldsymbol{a}_j^\dagger(t')\rangle\!\rangle.
\end{aligned} \tag{38}$$

Here the final term is responsible for correlations, and to process it we take its equation of motion on the right,

$$-\mathrm{i}\partial_{t'} \langle\!\langle \boldsymbol{b}_i(t)|\boldsymbol{a}_j^\dagger(t')\rangle\!\rangle = \delta(t - t') \langle \boldsymbol{b}_i|\boldsymbol{a}_j^\dagger\rangle + \sum_l \langle\!\langle \boldsymbol{b}_i(t)|\boldsymbol{a}_l^\dagger(t')\rangle\!\rangle (\tilde{\varepsilon}_{lj} - \mu\delta_{lj}) + \langle\!\langle \boldsymbol{b}_i(t)|\boldsymbol{b}_j^\dagger(t')\rangle\!\rangle. \tag{39}$$

Upon Fourier transforming, these two equations become

$$\left(\omega + \mu - \tilde{\varepsilon}_p\right) \langle\!\langle \boldsymbol{a}_p|\boldsymbol{a}_p^\dagger\rangle\!\rangle_\omega = \langle \boldsymbol{a}_p|\boldsymbol{a}_p^\dagger\rangle + \langle\!\langle \boldsymbol{b}_p|\boldsymbol{a}_p^\dagger\rangle\!\rangle_\omega, \tag{40}$$

$$\langle\!\langle \boldsymbol{b}_p|\boldsymbol{a}_p^\dagger\rangle\!\rangle_\omega \left(\omega + \mu - \tilde{\varepsilon}_p\right) = \langle \boldsymbol{b}_p|\boldsymbol{a}_p^\dagger\rangle + \langle\!\langle \boldsymbol{b}_p|\boldsymbol{b}_p^\dagger\rangle\!\rangle_\omega, \tag{41}$$

and combining them results in

$$\langle\!\langle \boldsymbol{a}_p|\boldsymbol{a}_p^\dagger\rangle\!\rangle_\omega = \frac{\langle \boldsymbol{a}_p|\boldsymbol{a}_p^\dagger\rangle}{\omega + \mu - \tilde{\varepsilon}_p} - \frac{1}{\omega + \mu - \tilde{\varepsilon}_p}\left(\langle \boldsymbol{b}_p|\boldsymbol{a}_p^\dagger\rangle + \langle\!\langle \boldsymbol{b}_p|\boldsymbol{b}_p^\dagger\rangle\!\rangle_\omega\right)\frac{1}{\omega + \mu - \tilde{\varepsilon}_p}. \tag{42}$$

This already takes an instructive form. Defining for convenience the correlators[2]

$$\begin{aligned}
I_{ij} &= \langle \boldsymbol{a}_i|\boldsymbol{a}_j^\dagger\rangle, \quad K_{ij} = \langle \boldsymbol{b}_i|\boldsymbol{a}_j^\dagger\rangle, \quad M_{ij}(t - t') = \langle\!\langle \boldsymbol{b}_i(t)|\boldsymbol{b}_j^\dagger(t')\rangle\!\rangle, \\
I_p &= \langle \boldsymbol{a}_p|\boldsymbol{a}_p^\dagger\rangle, \quad K_p = \langle \boldsymbol{b}_p|\boldsymbol{a}_p^\dagger\rangle, \quad M_p(\omega) = \langle\!\langle \boldsymbol{b}_p|\boldsymbol{b}_p^\dagger\rangle\!\rangle_\omega,
\end{aligned} \tag{43}$$

we may cast Eq. (42) as

$$G_p(\omega) = G_{0,p}(\omega) + G_{0,p}(\omega)T_p(\omega)G_{0,p}(\omega), \tag{44}$$

---

[2]We remark that $I_{ij} = \delta_{ij}(1 - \langle \boldsymbol{n}_i\rangle)$, and $\langle \boldsymbol{n}_i\rangle$ is independent of $i$ due to translational invariance. Consequently $I_p$ is independent of $p$. It is worthwhile however to maintain site and momentum dependence as it reflects that $I_p$ has a matrix structure when working with DOFs more complex than that of our model system. Indeed, for the case of multi-species DOFs we may regard the site index as a compound index which also denotes the species.

with bare Green's function

$$G_{0,p}(\omega) = \frac{I_p}{\omega + \mu - \tilde{\varepsilon}_p}, \tag{45}$$

and scattering matrix

$$T_p(\omega) = I_p^{-1}\big(K_p + M_p(\omega)\big)I_p^{-1}. \tag{46}$$

## 4.1 Dyson form

We now recast Eq. (44) as a Dyson equation

$$G_p(\omega) = G_{0,p}(\omega) + G_{0,p}(\omega)\Sigma_p(\omega)G_p(\omega). \tag{47}$$

Formally, this is immediately achieved upon introducing a self-energy through

$$T_p(\omega) = \Sigma_p(\omega) + \Sigma_p(\omega)G_{0,p}(\omega)T_p(\omega). \tag{48}$$

In practice, we can obtain a useful explicit expression for $\Sigma_p(\omega)$ following Tserkovnikov [17]. It is worthwhile to present the derivation, to see which information is employed.

First we re-express the left equation of motion Eq. (40) above as

$$\Big(\omega + \mu - \tilde{\varepsilon}_p - \langle\!\langle \boldsymbol{b}_p|\boldsymbol{a}_p^\dagger\rangle\!\rangle_\omega \langle\!\langle \boldsymbol{a}_p|\boldsymbol{a}_p^\dagger\rangle\!\rangle_\omega^{-1}\Big)\langle\!\langle \boldsymbol{a}_p|\boldsymbol{a}_p^\dagger\rangle\!\rangle_\omega = \langle \boldsymbol{a}_p|\boldsymbol{a}_p^\dagger\rangle, \tag{49}$$

and similarly the corresponding right equation of motion as

$$\langle\!\langle \boldsymbol{a}_p|\boldsymbol{a}_p^\dagger\rangle\!\rangle_\omega \Big(\omega + \mu - \tilde{\varepsilon}_p - \langle\!\langle \boldsymbol{a}_p|\boldsymbol{a}_p^\dagger\rangle\!\rangle_\omega^{-1} \langle\!\langle \boldsymbol{a}_p|\boldsymbol{b}_p^\dagger\rangle\!\rangle_\omega\Big) = \langle \boldsymbol{a}_p|\boldsymbol{a}_p^\dagger\rangle. \tag{50}$$

Extracting $\langle\!\langle \boldsymbol{a}_p|\boldsymbol{a}_p^\dagger\rangle\!\rangle_\omega^{-1}$ from the later and multiplying by $\langle\!\langle \boldsymbol{b}_p|\boldsymbol{a}_p^\dagger\rangle\!\rangle_\omega$, we obtain

$$\begin{aligned}
\langle\!\langle \boldsymbol{b}_p|\boldsymbol{a}_p^\dagger\rangle\!\rangle_\omega \langle\!\langle \boldsymbol{a}_p|\boldsymbol{a}_p^\dagger\rangle\!\rangle_\omega^{-1} &= \Big( \langle\!\langle \boldsymbol{b}_p|\boldsymbol{a}_p^\dagger\rangle\!\rangle_\omega \big(\omega + \mu - \tilde{\varepsilon}_p\big) - \langle\!\langle \boldsymbol{b}_p|\boldsymbol{a}_p^\dagger\rangle\!\rangle_\omega \langle\!\langle \boldsymbol{a}_p|\boldsymbol{a}_p^\dagger\rangle\!\rangle_\omega^{-1} \langle\!\langle \boldsymbol{a}_p|\boldsymbol{b}_p^\dagger\rangle\!\rangle_\omega \Big)\langle \boldsymbol{a}_p|\boldsymbol{a}_p^\dagger\rangle^{-1} \\
&= \Big( \langle \boldsymbol{b}_p|\boldsymbol{a}_p^\dagger\rangle + \langle\!\langle \boldsymbol{b}_p|\boldsymbol{b}_p^\dagger\rangle\!\rangle_\omega - \langle\!\langle \boldsymbol{b}_p|\boldsymbol{a}_p^\dagger\rangle\!\rangle_\omega \langle\!\langle \boldsymbol{a}_p|\boldsymbol{a}_p^\dagger\rangle\!\rangle_\omega^{-1} \langle\!\langle \boldsymbol{a}_p|\boldsymbol{b}_p^\dagger\rangle\!\rangle_\omega \Big)\langle \boldsymbol{a}_p|\boldsymbol{a}_p^\dagger\rangle^{-1},
\end{aligned} \tag{51}$$

with the second line following from Eq. (41). Substituting this back into Eq. (49) results in

$$\big((\omega + \mu)I_p - \tilde{\varepsilon}_p I_p - K_p - M_p^\star(\omega)\big)I_p^{-1}G_p(\omega) = I_p, \tag{52}$$

and so we obtain that $G_p(\omega)$ takes the Dyson form of Eq. (47) with self-energy

$$\Sigma_p(\omega) = I_p^{-1}\big(K_p + M_p^\star(\omega)\big)I_p^{-1}, \tag{53}$$

where the irreducible $M_p^\star(\omega)$ is given by

$$M_p^\star(\omega) = \langle\!\langle \boldsymbol{b}_p|\boldsymbol{b}_p^\dagger\rangle\!\rangle_\omega^\star = \langle\!\langle \boldsymbol{b}_p|\boldsymbol{b}_p^\dagger\rangle\!\rangle_\omega - \langle\!\langle \boldsymbol{b}_p|\boldsymbol{a}_p^\dagger\rangle\!\rangle_\omega \langle\!\langle \boldsymbol{a}_p|\boldsymbol{a}_p^\dagger\rangle\!\rangle_\omega^{-1} \langle\!\langle \boldsymbol{a}_p|\boldsymbol{b}_p^\dagger\rangle\!\rangle_\omega, \tag{54}$$

or explicitly in space and time as

$$M_{ij}^{\star}(t-t') = V_{i\boldsymbol{k}}V_{\boldsymbol{l}j} \langle\!\langle \boldsymbol{n_k}(t)\boldsymbol{a}_i(t)|\boldsymbol{a}_j^{\dagger}(t')\boldsymbol{n_l}(t')\rangle\!\rangle^{\star} + \lambda^2\varepsilon_{i\boldsymbol{k}}\varepsilon_{\boldsymbol{l}j} \langle\!\langle \boldsymbol{n}_i(t)\boldsymbol{a_k}(t)|\boldsymbol{a_l}^{\dagger}(t')\boldsymbol{n}_j(t')\rangle\!\rangle^{\star}$$
$$- \lambda V_{i\boldsymbol{k}}\varepsilon_{\boldsymbol{l}j} \langle\!\langle \boldsymbol{n_k}(t)\boldsymbol{a}_i(t)|\boldsymbol{a_l}^{\dagger}(t')\boldsymbol{n}_j(t')\rangle\!\rangle^{\star} - \lambda\varepsilon_{i\boldsymbol{k}}V_{\boldsymbol{l}j} \langle\!\langle \boldsymbol{n}_i(t)\boldsymbol{a_k}(t)|\boldsymbol{a}_j^{\dagger}(t')\boldsymbol{n_l}(t')\rangle\!\rangle^{\star},$$

$$(55)$$

with site indices in bold font denoting summation over all sites.

This final object $M_p^{\star}(\omega)$ is universal in the sense that it is insensitive to how $\tilde{\varepsilon}_{ij}$ is chosen in Eq. (37). We can replace $\boldsymbol{b}_i \to \mathrm{i}\dot{\boldsymbol{a}}_i$, $\boldsymbol{b}_i^{\dagger} \to -\mathrm{i}\dot{\boldsymbol{a}}_i^{\dagger}$ in Eq. (54) and not alter the expression. The combination

$$K_p^{\star} = \tilde{\varepsilon}_p I_p + K_p, \tag{56}$$

is also independent of how $\tilde{\varepsilon}_{ij}$ is chosen. Indeed we can express this equivalently as $K_{ij}^{\star} = \langle \mathrm{i}\dot{\boldsymbol{a}}_i + \mu\boldsymbol{a}_i|\boldsymbol{a}_j^{\dagger}\rangle$, which is given explicitly by

$$K_{ij}^{\star} = \big(1 - \lambda\langle\boldsymbol{n}_i\rangle\big)\varepsilon_{ij}\big(1 - \lambda\langle\boldsymbol{n}_j\rangle\big) - \lambda\delta_{ij}\sum_l \varepsilon_{il}\langle\boldsymbol{a}_l\boldsymbol{a}_i^{\dagger}\rangle + \lambda^2\varepsilon_{ij}\big(\langle\boldsymbol{n}_i\boldsymbol{n}_j\rangle - \langle\boldsymbol{n}_i\rangle\langle\boldsymbol{n}_j\rangle\big)$$
$$+ \delta_{ij}\big(1 - \lambda\langle\boldsymbol{n}_i\rangle\big)\sum_l V_{il}\langle\boldsymbol{n}_l\rangle + V_{ij}\langle\boldsymbol{a}_i\boldsymbol{a}_j^{\dagger}\rangle - \lambda\delta_{ij}\sum_l V_{il}\big(\langle\boldsymbol{n}_i\boldsymbol{n}_l\rangle - \langle\boldsymbol{n}_i\rangle\langle\boldsymbol{n}_l\rangle\big).$$

$$(57)$$

To summarise this formal analysis it is instructive to cast the Dyson equation in the form

$$G_p(\omega) = I_p\frac{1}{(\omega + \mu)I_p - K_p^{\star} - M_p^{\star}(\omega)}I_p. \tag{58}$$

Let us analyse the various contributions:

- The three factors of $I_p$ account for the non-orthogonality of non-canonical DOFs, in contrast to a canonical DOF for which $I_{ij} = \delta_{ij}$. The factors of $I_p$ at each side encode a non-trivial overlap onto the propagator in between. This becomes particularly transparent if one considers a multi-species non-canonical DOF, where the Green's function of one particle type may be influenced by correlations induced by the propagation of other particle types.

- The term $K_p^{\star}$ gives the mean-field single-particle modes, encoding static correlations. That is, neglecting $M_p^{\star}(\omega)$ gives a generalisation of the canonical Hartree–Fock approximation (which is reobtained at $\lambda = 0$).

- The term $M_p^{\star}(\omega)$ encodes dynamical correlations. These are orthogonal to the mean-field single-particle modes, and characterise their dressing as quasi-particles. The task of computing $G_p(\omega)$ has thus been transferred to the task of computing $M_p^{\star}(\omega)$. In principle one can recursively apply the analysis to $M_p^{\star}(\omega)$, thereby generating a hierarchical structure successively organising correlations in orders of orthogonality [15,17]. In practice one may seek to find an adequate approximation for $M_p^{\star}(\omega)$. The most commonly employed approximation is the mode-coupling approximation [2,29], which decouples the density-density correlations in Eq. (55) (see Eq. (105) below). This indeed leads to well-defined quasi-particles, which are long-lived at low energy and low temperature, as follows generally from Landau's phase space argument [57].

For a closely related discussion, albeit at the level of a hydrodynamic description, see Sec. 5.6 of Ref. [58].

# 5   Organising correlations – the Schwinger method

We now turn to the Schwinger method [1,37], which offers a complementary means of organising the correlations of our model system. We mirror our analysis with the previous section, and in doing so we reobtain the Dyson form of Eq. (58). This leads us to a closed expression for $M_p^\star(\omega)$ taking a functional differential form. We discuss issues related to generating approximations consistent with conservation laws. We then analyse a resummation of density correlations and derive an approximation capturing screening.

We remark that our analysis here differs from a related approach developed by Shastry [40–42], employed also in [4, 43], where a factorisation ansatz in employed to cast the Green's function through two self-energy-like objects. This approach also faces issues in generating conserving approximations.

## 5.1   Imaginary-time formalism

The Schwinger method employs the imaginary-time formalism, where $\mathcal{O}(\tau) = e^{\tau \boldsymbol{H}} \mathcal{O} e^{-\tau \boldsymbol{H}}$. We consider here $\tau$-ordered correlation functions in the presence of an external source $\mathcal{U}$ as follows

$$\langle \mathcal{O}(\tau_1, \tau_2, \dots) \rangle = \frac{\text{Tr}\left( e^{-\beta \boldsymbol{H}} \mathcal{T}\left[ \mathcal{U} \mathcal{O}(\tau_1, \tau_2, \dots) \right] \right)}{\text{Tr}\left( e^{-\beta \boldsymbol{H}} \mathcal{T}[\mathcal{U}] \right)}, \tag{59}$$

where $\mathcal{T}$ is the $\tau$-ordering operator $\mathcal{T}\left[ \mathcal{O}(\tau) \mathcal{O}'(\tau') \right] = \vartheta(\tau - \tau') \mathcal{O}(\tau) \mathcal{O}'(\tau') \pm \vartheta(\tau' - \tau) \mathcal{O}'(\tau') \mathcal{O}(\tau)$. We will focus on inhomogeneous sources coupling to the local density,

$$\mathcal{U} = \exp\Big( \sum_i \int_0^\beta \mathrm{d}\tau\, \zeta_i(\tau) \boldsymbol{n}_i(\tau) \Big). \tag{60}$$

Denoting variations with respect to the sources as $\nabla_i(\tau) = \frac{\delta}{\delta \zeta_i(\tau^+)}$, we then have that

$$\nabla_i(\tau) \langle \mathcal{O}(\tau_1, \tau_2, \dots, \tau_n) \rangle = \langle \boldsymbol{n}_i(\tau) \mathcal{O}(\tau_1, \tau_2, \dots, \tau_n) \rangle - \langle \boldsymbol{n}_i(\tau) \rangle \langle \mathcal{O}(\tau_1, \tau_2, \dots, \tau_n) \rangle. \tag{61}$$

(Here $\tau^+ = \tau + 0^+$ incorporates an infinitesimal regulator which ensures a consistent ordering when $\tau$ is one of the $\tau_1, \tau_2, \dots, \tau_n$. In the following we will suppress this when unimportant.)

Our primary object of interest is the Green's function

$$\mathcal{G}_{ij}(\tau, \tau') = - \langle \boldsymbol{a}_i(\tau) \boldsymbol{a}_j^\dagger(\tau') \rangle. \tag{62}$$

This is fixed through its imaginary-time equation of motion, combined with the KMS boundary condition

$$\mathcal{G}_{ij}(\beta, \tau) = \pm \mathcal{G}_{ij}(0, \tau), \tag{63}$$

which follows from the cyclicity of the trace.

The Schwinger method provides a means of generating systematic approximations for $\mathcal{G}_{ij}(\tau, \tau')$. Once an approximation is identified the external sources can be set to zero, restoring space and $\tau$ translational invariance. From the KMS boundary condition, the Fourier transform

$$\mathcal{G}_p(\mathrm{i}\omega_n) = \tfrac{1}{V} \sum_{i,j} \int_0^\beta \mathrm{d}(\tau - \tau')\, e^{\mathrm{i}\omega_n(\tau-\tau')-\mathrm{i}p(i-j)} \mathcal{G}_{ij}(\tau, \tau'), \tag{64}$$

is defined at the Matsubara frequencies $\omega_n = (2n + 1/2 \mp 1/2)\frac{\pi}{\beta}$ with $n \in \mathbb{Z}$. Provided the resulting approximation is manifestly causal, we can in principle then analytically continue $\mathcal{G}_p(\mathrm{i}\omega_n)$ to all $\omega$ away from the real axis, and as a result obtain the corresponding retarded Green's function as

$$G_p(\omega) = \mathcal{G}_p(\omega + \mathrm{i}0^+). \tag{65}$$

## 5.2   Paired equations of motion

We proceed to analyse $\mathcal{G}_{ij}(\tau, \tau')$ through its equation of motion. Let us consider simultaneously both the left and right equations of motion, as in the derivation of the Dyson form of the retarded Green's function from Eqs. (49) and (50) in Sec. 4. Here we have[3]

$$\begin{aligned}
-\partial_\tau \mathcal{G}_{ij}(\tau, \tau') + \zeta_i(\tau)\mathcal{G}_{ij}(\tau, \tau') - \langle [\boldsymbol{H}, \boldsymbol{a}_i(\tau)]\boldsymbol{a}_j^\dagger(\tau')\rangle = \mathcal{I}_{ij}(\tau, \tau'), \\
\partial_{\tau'} \mathcal{G}_{ij}(\tau, \tau') + \mathcal{G}_{ij}(\tau, \tau')\zeta_j(\tau') + \langle \boldsymbol{a}_i(\tau)[\boldsymbol{H}, \boldsymbol{a}_j^\dagger(\tau')]\rangle = \mathcal{I}_{ij}(\tau, \tau'),
\end{aligned} \tag{67}$$

with

$$\mathcal{I}_{ij}(\tau, \tau') = \delta(\tau - \tau')\delta_{ij}\big(1 - \lambda\langle \boldsymbol{n}_i(\tau)\rangle\big). \tag{68}$$

Evaluating the commutators from Eq. (36), and decoupling the correlations via Eq. (61), we re-express the left-hand sides of Eqs. (67) as

$$\left[\delta_{i\boldsymbol{k}}\big(-\partial_\tau + \mu + \zeta_i(\tau)\big) - \varepsilon_{i\boldsymbol{k}} + \lambda\varepsilon_{i\boldsymbol{k}}\big(\langle \boldsymbol{n}_i(\tau)\rangle + \nabla_i(\tau)\big) - \delta_{i\boldsymbol{k}}V_{i\boldsymbol{l}}\big(\langle \boldsymbol{n}_{\boldsymbol{l}}(\tau)\rangle + \nabla_{\boldsymbol{l}}(\tau)\big)\right]\mathcal{G}_{\boldsymbol{k}j}(\tau, \tau'),$$

$$\left[\delta_{\boldsymbol{k}j}\big(\partial_{\tau'} + \mu + \zeta_j(\tau')\big) - \varepsilon_{\boldsymbol{k}j} + \lambda\varepsilon_{\boldsymbol{k}j}\big(\langle \boldsymbol{n}_j(\tau')\rangle + \nabla_j(\tau')\big) - \delta_{\boldsymbol{k}j}V_{\boldsymbol{l}j}\big(\langle \boldsymbol{n}_{\boldsymbol{l}}(\tau')\rangle + \nabla_{\boldsymbol{l}}(\tau')\big)\right]\mathcal{G}_{i\boldsymbol{k}}(\tau, \tau'). \tag{69}$$

Here site indices in bold are summed over all sites, and we will further employ $\tau$ in bold to denote a variable integrated from 0 to $\beta$. Employing the identity $\nabla\mathcal{G} = -\mathcal{G}(\nabla\mathcal{G}^{-1})\mathcal{G}$, we then cast the

---

[3] The contributions of both $\mathcal{I}$ on the right-hand side and the source $\zeta$ on the left-hand side are consequences of the $\tau$-ordering operator $\mathcal{T}$. For the latter, the $\tau$ dependence can be seen for example as follows

$$\mathcal{G}_{ij}(\tau, \tau') = -\frac{\mathrm{Tr}\left(e^{-\beta\boldsymbol{H}}\mathcal{T}[e^{\sum_i \int_\tau^\beta \mathrm{d}\tilde\tau\, \zeta_i(\tilde\tau)\boldsymbol{n}_i(\tilde\tau)}\boldsymbol{a}_i(\tau)e^{\sum_i \int_0^\tau \mathrm{d}\tilde\tau\, \zeta_i(\tilde\tau)\boldsymbol{n}_i(\tilde\tau)}\boldsymbol{a}_j^\dagger(\tau')]\right)}{\mathrm{Tr}\left(e^{-\beta\boldsymbol{H}}\mathcal{T}[e^{\sum_i \int_0^\beta \mathrm{d}\tilde\tau\, \zeta_i(\tilde\tau)\boldsymbol{n}_i(\tilde\tau)}]\right)}. \tag{66}$$

equations of motion compactly as

$$\mathcal{D}_{il}(\tau,\tilde{\boldsymbol{\tau}})\mathcal{G}_{lj}(\tilde{\boldsymbol{\tau}},\tau') = \mathcal{I}_{ij}(\tau,\tau'),$$
$$\mathcal{G}_{il}(\tau,\tilde{\boldsymbol{\tau}})\overline{\mathcal{D}}_{lj}(\tilde{\boldsymbol{\tau}},\tau') = \mathcal{I}_{ij}(\tau,\tau'),$$

(70)

upon defining

$$\mathcal{D}_{ij}(\tau,\tau') = (-\partial_\tau + \mu)\delta(\tau-\tau')\delta_{ij} + \delta(\tau-\tau')\delta_{ij}\zeta_i(\tau) - \mathcal{F}_{ij}(\tau,\tau'),$$
$$\overline{\mathcal{D}}_{ij}(\tau,\tau') = (-\partial_\tau + \mu)\delta(\tau-\tau')\delta_{ij} + \delta(\tau-\tau')\delta_{ij}\zeta_i(\tau) - \overline{\mathcal{F}}_{ij}(\tau,\tau'),$$

(71)

with

$$\mathcal{F}_{ij}(\tau,\tau') = \delta(\tau-\tau')\big(1 - \lambda\langle\boldsymbol{n}_i(\tau)\rangle\big)\varepsilon_{ij} + \lambda\varepsilon_{il}\mathcal{G}_{lk}(\tau,\tilde{\boldsymbol{\tau}})\nabla_i(\tau)\mathcal{G}_{kj}^{-1}(\tilde{\boldsymbol{\tau}},\tau')$$
$$+ \delta(\tau-\tau')\delta_{ij}V_{il}\langle\boldsymbol{n}_l(\tau)\rangle - V_{il}\mathcal{G}_{ik}(\tau,\tilde{\boldsymbol{\tau}})\nabla_l(\tau)\mathcal{G}_{kj}^{-1}(\tilde{\boldsymbol{\tau}},\tau'),$$
$$\overline{\mathcal{F}}_{ij}(\tau,\tau') = \delta(\tau-\tau')\varepsilon_{ij}\big(1 - \lambda\langle\boldsymbol{n}_j(\tau)\rangle\big) + \lambda\big(\nabla_j(\tau')\mathcal{G}_{ik}^{-1}(\tau,\tilde{\boldsymbol{\tau}})\big)\mathcal{G}_{kl}(\tilde{\boldsymbol{\tau}},\tau')\varepsilon_{lj}$$
$$+ \delta(\tau-\tau')\delta_{ij}\langle\boldsymbol{n}_l(\tau)\rangle V_{li} - \big(\nabla_l(\tau')\mathcal{G}_{ik}^{-1}(\tau,\tilde{\boldsymbol{\tau}})\big)\mathcal{G}_{kj}(\tilde{\boldsymbol{\tau}},\tau')V_{lj}.$$

(72)

Here we have introduced a notation of putting a line over objects originating from the right equation of motion.

We now combine the left and right equations to establish a direct equivalence with the analysis of Sec. 4. To proceed we can either evaluate $\mathcal{F}$ using $\mathcal{G}^{-1} = \overline{\mathcal{D}}\mathcal{I}^{-1}$, i.e. that

$$\mathcal{G}_{ik}(\tau,\tilde{\boldsymbol{\tau}})\nabla_l(\tau'')\mathcal{G}_{kj}^{-1}(\tilde{\boldsymbol{\tau}},\tau') = \mathcal{G}_{il}(\tau,\tau'')\mathcal{I}_{lj}^{-1}(\tau'',\tau') - \mathcal{G}_{ik}(\tau,\tilde{\boldsymbol{\tau}})\big(\nabla_l(\tau'')\overline{\mathcal{F}}_{kk'}(\tilde{\boldsymbol{\tau}},\tilde{\boldsymbol{\tau}}')\big)\mathcal{I}_{k'j}^{-1}(\tilde{\boldsymbol{\tau}}',\tau')$$
$$+ \lambda\big(\nabla_l(\tau'')\langle\boldsymbol{n}_i(\tau)\rangle\big)\mathcal{I}_{ij}^{-1}(\tau,\tau'),$$

(73)

where for the final term we use $\mathcal{G}\overline{\mathcal{D}} = \mathcal{I}$ and $\mathcal{I}\nabla\mathcal{I}^{-1} = -(\nabla\mathcal{I})\mathcal{I}^{-1} = \lambda(\nabla\langle\boldsymbol{n}\rangle)\mathcal{I}^{-1}$, or alternatively evaluate $\overline{\mathcal{F}}$ using $\mathcal{G}^{-1} = \mathcal{I}^{-1}\mathcal{D}$, i.e. that

$$\big(\nabla_l(\tau'')\mathcal{G}_{ik}^{-1}(\tau,\tilde{\boldsymbol{\tau}})\big)\mathcal{G}_{kj}(\tilde{\boldsymbol{\tau}},\tau') = \mathcal{I}_{il}^{-1}(\tau,\tau'')\mathcal{G}_{lj}(\tau'',\tau') - \mathcal{I}_{ik}^{-1}(\tau,\tilde{\boldsymbol{\tau}})\big(\nabla_l(\tau'')\mathcal{F}_{kk'}(\tilde{\boldsymbol{\tau}},\tilde{\boldsymbol{\tau}}')\big)\mathcal{G}_{k'j}(\tilde{\boldsymbol{\tau}}',\tau')$$
$$+ \lambda\mathcal{I}_{ij}^{-1}(\tau,\tau')\big(\nabla_l(\tau'')\langle\boldsymbol{n}_j(\tau')\rangle\big).$$

(74)

Focusing first on the left equation of motion, we can thus express $\mathcal{F}$ as

$$\mathcal{F}_{ij}(\tau,\tau') = \mathcal{K}_{il}^\star(\tau)\mathcal{I}_{lj}^{-1}(\tau,\tau') + \mathcal{M}_{il}^\star(\tau,\tilde{\boldsymbol{\tau}})\mathcal{I}_{lj}^{-1}(\tilde{\boldsymbol{\tau}},\tau')$$

(75)

where

$$\mathcal{K}_{ij}^\star(\tau) = \big(1 - \lambda\langle\boldsymbol{n}_i(\tau)\rangle\big)\varepsilon_{ij}\big(1 - \lambda\langle\boldsymbol{n}_j(\tau)\rangle\big) + \lambda\delta_{ij}\varepsilon_{il}\mathcal{G}_{li}(\tau,\tau) + \lambda^2\varepsilon_{ij}\nabla_i(\tau)\langle\boldsymbol{n}_j(\tau)\rangle$$
$$+ \delta_{ij}\big(1 - \lambda\langle\boldsymbol{n}_i(\tau)\rangle\big)V_{il}\langle\boldsymbol{n}_l(\tau)\rangle - V_{ij}\mathcal{G}_{ij}(\tau,\tau) - \lambda\delta_{ij}V_{il}\nabla_l(\tau)\langle\boldsymbol{n}_i(\tau)\rangle,$$

(76)

which reduces to $K_{ij}^\star$ of Eq. (57) upon switching off the sources, and

$$\mathcal{M}_{ij}^\star(\tau,\tau') = V_{il}\mathcal{G}_{ik}(\tau,\tilde{\boldsymbol{\tau}})\nabla_l(\tau)\overline{\mathcal{F}}_{kj}(\tilde{\boldsymbol{\tau}},\tau') - \lambda\varepsilon_{il}\mathcal{G}_{lk}(\tau,\tilde{\boldsymbol{\tau}})\nabla_i(\tau)\overline{\mathcal{F}}_{kj}(\tilde{\boldsymbol{\tau}},\tau').$$

(77)

In this way we have cast the left equation of motion in the form

$$\left[\left(-\partial_\tau + \zeta_i(\tau)\right)\mathcal{I}_{il}(\tau, \tilde{\boldsymbol{\tau}}) - \mathcal{K}_{il}^\star(\tau)\delta(\tau - \tilde{\boldsymbol{\tau}}) - \mathcal{M}_{il}^\star(\tau, \tilde{\boldsymbol{\tau}})\right]\mathcal{I}_{lk}^{-1}(\tilde{\boldsymbol{\tau}}, \tilde{\boldsymbol{\tau}}')\mathcal{G}_{kj}(\tilde{\boldsymbol{\tau}}', \tau') = \mathcal{I}_{ij}(\tau, \tau'), \quad (78)$$

which is a direct analogue of Eq. (52) above. As a consequence it follows that $\mathcal{M}_p^\star(\omega + \mathrm{i}0^+)$ is precisely equivalent to $M_p^\star(\omega)$ of Eq. (55) in the zero source limit, and a straightforward yet lengthy manipulation of the $\nabla\overline{\mathcal{F}}$ terms in Eq. (77) from Eq. (72) indeed demonstrates that this is so.

Similarly the right equation of motion takes the form

$$\mathcal{G}_{ik}(\tau, \tilde{\boldsymbol{\tau}})\mathcal{I}_{kl}^{-1}(\tilde{\boldsymbol{\tau}}, \tilde{\boldsymbol{\tau}}')\left[\left(\partial_{\tau'} + \zeta_j(\tau')\right)\mathcal{I}_{lj}(\tilde{\boldsymbol{\tau}}', \tau') - \delta(\tilde{\boldsymbol{\tau}}' - \tau')\mathcal{K}_{lj}^\star(\tau') - \mathcal{M}_{lj}^\star(\tilde{\boldsymbol{\tau}}', \tau')\right] = \mathcal{I}_{ij}(\tau, \tau'), \quad (79)$$

where here we use Eq. (74) to express

$$\overline{\mathcal{F}}_{ij}(\tau, \tau') = \mathcal{I}_{il}^{-1}(\tau, \tau')\overline{\mathcal{K}}_{lj}^\star(\tau') + \mathcal{I}_{il}^{-1}(\tau, \tilde{\boldsymbol{\tau}})\overline{\mathcal{M}}_{lj}^\star(\tilde{\boldsymbol{\tau}}, \tau'), \quad (80)$$

with

$$\begin{aligned}
\overline{\mathcal{K}}_{ij}^\star(\tau) = &\left(1 - \lambda\langle\boldsymbol{n}_i(\tau)\rangle\right)\varepsilon_{ij}\left(1 - \lambda\langle\boldsymbol{n}_j(\tau)\rangle\right) + \lambda\delta_{ij}\mathcal{G}_{il}(\tau, \tau)\varepsilon_{li} + \lambda^2\varepsilon_{ij}\nabla_j(\tau)\langle\boldsymbol{n}_i(\tau)\rangle \\
&+ \delta_{ij}\left(1 - \lambda\langle\boldsymbol{n}_i(\tau)\rangle\right)V_{li}\langle\boldsymbol{n}_l(\tau)\rangle - V_{ij}\mathcal{G}_{ij}(\tau, \tau) - \lambda\delta_{ij}V_{li}\nabla_l(\tau)\langle\boldsymbol{n}_i(\tau)\rangle,
\end{aligned} \quad (81)$$

and

$$\overline{\mathcal{M}}_{ij}^\star(\tau, \tau') = V_{lj}\left(\nabla_l(\tau')\mathcal{F}_{ik}(\tau, \tilde{\boldsymbol{\tau}})\right)\mathcal{G}_{kj}(\tilde{\boldsymbol{\tau}}, \tau') - \lambda\varepsilon_{lj}\left(\nabla_j(\tau')\mathcal{F}_{ik}(\tau, \tilde{\boldsymbol{\tau}})\right)\mathcal{G}_{kl}(\tilde{\boldsymbol{\tau}}, \tau'). \quad (82)$$

We have $\overline{\mathcal{K}}_{ij}(\tau) = \overline{\mathcal{K}}_{ij}^\star(\tau)$ as $\nabla_i(\tau)\langle\boldsymbol{n}_j(\tau)\rangle = \nabla_j(\tau)\langle\boldsymbol{n}_i(\tau)\rangle = \langle\boldsymbol{n}_i(\tau)\boldsymbol{n}_j(\tau)\rangle - \langle\boldsymbol{n}_i(\tau)\rangle\langle\boldsymbol{n}_j(\tau)\rangle$ is the static susceptibility, and using symmetries of $\varepsilon_{ij}$ and $V_{ij}$. Again $\overline{\mathcal{M}}_p^\star(\omega + \mathrm{i}0^+)$ can be explicitly seen to be equivalent to $M_p^\star(\omega)$ in the zero source limit.

This paired set of functional differential equations, Eqs. (75)-(77) and Eqs. (80)-(82), thus provide exact closed expressions for $K_p^\star$ and $M_p^\star(\omega)$ appearing in Eq. (58), framed in terms of the unknown $\mathcal{G}_p(\omega)$. There are no known methods for solving such equations in general however. Instead we may proceed as is done in the canonical case by evaluating the above functional derivatives in a perturbative manner, yielding self-consistent equations for $\mathcal{G}_p(\omega)$, and thereby organising the correlations around the underlying DOF.

As we cast our analysis here largely in parallel with that of Sec. 4, let us emphasise the difference. Previously Eq. (52) was obtained by treating correlated terms through their equations of motion, whereas here Eq. (78) is obtained by relating the correlated terms to variations of the external sources via Eq. (61). The advantage of the present approach is that we can manipulate the functional derivative, providing a systematic means of organising correlations. Moreover, the Schwinger approach provides a powerful framework upon which resummations of correlations can be formulated. We return to this below in Sec. 5.4 where we obtain an approximation which captures the screening of density correlations.

## 5.3 Conserving approximations

Key to our derivation here has been to analyse the left and right equations of motion in tandem. This enabled us to identify $\mathcal{M}^\star$, or equivalently $\overline{\mathcal{M}}^\star$, related to the Dyson form of Sec. 4. Ultimately the purpose of these equations is to generate approximations for the Green's function, and the question arises whether results obtained from Eqs. (77) and (82) are consistent. From Eq. (70) we have that $\mathcal{DI} = \mathcal{ID}$, and consequently we should require that any approximate computation should yield a result satisfying the consistency condition

$$\mathcal{M}_{ij}^\star(\tau,\tau') \stackrel{?}{=} \overline{\mathcal{M}}_{ij}^\star(\tau,\tau'). \tag{83}$$

This is an important issue. Consistency guarantees that an approximation is manifestly causal, thus permitting access to the retarded Green's function through Eq. (65). Moreover, that Eq. (83) is obeyed in the presence of external sources is the key condition for having a conserving approximation in the sense of Baym–Kadanoff [1,59], whose analysis carries over to the non-canonical case largely unchanged. That is, it is essential for ensuring that an approximation respects conservation laws for conserved charges, as well as for momentum and energy, and is thus capable of adequately describing transport phenomena.

This is also a delicate issue. To demonstrate, let us note that any effort to compute and equate $\mathcal{M}^\star$ and $\overline{\mathcal{M}}^\star$ gives rise to corresponding terms of the form

$$\nabla_i(\tau) \langle \boldsymbol{n}_j(\tau') \rangle \stackrel{?}{=} \nabla_j(\tau') \langle \boldsymbol{n}_i(\tau) \rangle. \tag{84}$$

Indeed the $\tau = \tau'$ version of this condition already appears upon equating $\mathcal{K}^\star$ of Eq. (76) with $\overline{\mathcal{K}}^\star$ of Eq. (81). Formally this equality is certainly true, as both sides equal the dynamical suscepti­bility $\chi_{ij}(\tau,\tau') = \langle \boldsymbol{n}_i(\tau)\boldsymbol{n}_j(\tau') \rangle - \langle \boldsymbol{n}_i(\tau) \rangle \langle \boldsymbol{n}_j(\tau') \rangle$. The question is whether this equality can be maintained upon evaluating the source derivatives, and ultimately taking an approximation. Let us first discuss the issues that arise, and we then comment upon possible routes towards a resolution.

To compute the source derivatives in Eq. (84) we relate the local expectation value of the density to the Green's function through $\langle \boldsymbol{n}_i(\tau) \rangle = \langle \boldsymbol{a}_i^\dagger(\tau)\boldsymbol{a}_i(\tau) \rangle = \mp \mathcal{G}_{ii}(\tau,\tau^+)$, which yields

$$\nabla_i(\tau) \langle \boldsymbol{n}_j(\tau') \rangle = \pm \mathcal{G}_{j\boldsymbol{k}}(\tau',\tilde{\boldsymbol{\tau}})\big(\nabla_i(\tau)\mathcal{G}_{\boldsymbol{k}l}^{-1}(\tilde{\boldsymbol{\tau}},\tilde{\boldsymbol{\tau}}')\big)\mathcal{G}_{lj}(\tilde{\boldsymbol{\tau}}',\tau'+0^+). \tag{85}$$

As in the previous subsection, we can evaluate $\nabla\mathcal{G}^{-1}$ using either $\mathcal{G}^{-1} = \overline{\mathcal{D}}\mathcal{I}^{-1}$ or $\mathcal{G}^{-1} = \mathcal{I}^{-1}\mathcal{D}$, i.e. Eq. (73) or Eq. (74). In the previous case there was a natural asymmetry between the left and right equations of motion, and it was unambiguous which form of $\mathcal{G}^{-1}$ to use in evaluating $\mathcal{F}$ and $\overline{\mathcal{F}}$ from Eq. (72). In contrast here Eq. (85) has a symmetric form, which now gets broken. Employing Eq. (73) gives

$$\begin{aligned}
\nabla_i(\tau) \langle \boldsymbol{n}_j(\tau') \rangle = {}&\pm\lambda\big(\nabla_i(\tau) \langle \boldsymbol{n}_j(\tau') \rangle\big)\mathcal{G}_{jj}(\tau',\tau'+0^+) \pm \mathcal{G}_{ji}(\tau',\tau)\mathcal{I}_{il}^{-1}(\tau,\tilde{\boldsymbol{\tau}})\mathcal{G}_{lj}(\tilde{\boldsymbol{\tau}},\tau') \\
&\mp \mathcal{G}_{j\boldsymbol{k}}(\tau',\tilde{\boldsymbol{\tau}})\big(\nabla_i(\tau)\overline{\mathcal{F}}_{\boldsymbol{k}\boldsymbol{k}'}(\tilde{\boldsymbol{\tau}},\tilde{\boldsymbol{\tau}}')\big)\mathcal{I}_{\boldsymbol{k}'l}^{-1}(\tilde{\boldsymbol{\tau}}',\tilde{\boldsymbol{\tau}}'')\mathcal{G}_{lj}(\tilde{\boldsymbol{\tau}}'',\tau'),
\end{aligned} \tag{86}$$

while employing Eq. (74) gives

$$
\begin{aligned}
\nabla_i(\tau)\langle \boldsymbol{n}_j(\tau')\rangle = {}&\pm\lambda\big(\nabla_i(\tau)\langle \boldsymbol{n}_j(\tau')\rangle\big)\mathcal{G}_{jj}(\tau',\tau'+0^+) \pm \mathcal{G}_{jl}(\tau',\tilde{\boldsymbol{\tau}})\mathcal{I}_{li}^{-1}(\tilde{\boldsymbol{\tau}},\tau)\mathcal{G}_{ij}(\tau,\tau') \\
&\mp \mathcal{G}_{jl}(\tau',\tilde{\boldsymbol{\tau}})\mathcal{I}_{lk}^{-1}(\tilde{\boldsymbol{\tau}},\tilde{\boldsymbol{\tau}}')\big(\nabla_i(\tau)\mathcal{F}_{kk'}(\tilde{\boldsymbol{\tau}}',\tilde{\boldsymbol{\tau}}'')\big)\mathcal{G}_{k'j}(\tilde{\boldsymbol{\tau}}'',\tau'),
\end{aligned}
\tag{87}
$$

The resulting expressions differ, but only slightly through the $(\nabla\overline{\mathcal{F}})\mathcal{I}^{-1}$ vs. $\mathcal{I}^{-1}(\nabla\mathcal{F})$ terms, recalling that $\mathcal{I}$ is given by Eq. (68). Neither expression however, nor a combination of them, are symmetric under $i \leftrightarrow j$, $\tau \leftrightarrow \tau'$, and thus the consistency condition of Eq. (84) is in general violated.

It is perhaps worth highlighting that for a canonical DOF this issue is bypassed. There one has $\lambda = 0$ and $\mathcal{I}_{ij}^{-1}(\tau,\tau') = \delta(\tau-\tau')\delta_{ij}$. Then perturbatively evaluating $\nabla\mathcal{F}$ or $\nabla\overline{\mathcal{F}}$ one does obtain symmetric expressions. Indeed it is consistency conditions, of which Eq. (84) is a special case, which underlie the Baym–Kadanoff construction of the canonical Luttinger–Ward functional [60]. We may view the difficulties faced here as an obstruction to a straightforward construction of an analogous functional for non-canonical DOFs.

One route forward is to switch attention to more general external sources of the form $\mathcal{U} = \exp\big(\sum_{i,j}\int_0^\beta \mathrm{d}\tau\,\mathrm{d}\tau'\,\zeta_{ij}(\tau,\tau')\boldsymbol{a}_i^\dagger(\tau)\boldsymbol{a}_j(\tau')\big)$. Indeed such source terms are useful for analysing vertex corrections, which play an important role in the microscopic derivation of a Fermi liquid. A subtlety here is that such sources induce correlations for a non-canonical DOF, i.e. Eqs. (67) become

$$
\begin{aligned}
-\partial_\tau \mathcal{G}_{ij}(\tau,\tau') + \zeta_{il}(\tau,\tilde{\boldsymbol{\tau}})\langle\big(1-\lambda\boldsymbol{n}_i(\tau)\big)\boldsymbol{a}_l(\tilde{\boldsymbol{\tau}})\boldsymbol{a}_j^\dagger(\tau')\rangle - \langle[\boldsymbol{H},\boldsymbol{a}_i(\tau)]\boldsymbol{a}_j^\dagger(\tau')\rangle &= \mathcal{I}_{ij}(\tau,\tau'), \\
\partial_{\tau'}\mathcal{G}_{ij}(\tau,\tau') + \langle\boldsymbol{a}_i(\tau)\boldsymbol{a}_l^\dagger(\tilde{\boldsymbol{\tau}})\big(1-\lambda\boldsymbol{n}_j(\tau')\big)\rangle\zeta_{lj}(\tilde{\boldsymbol{\tau}},\tau') + \langle\boldsymbol{a}_i(\tau)[\boldsymbol{H},\boldsymbol{a}_j^\dagger(\tau')]\rangle &= \mathcal{I}_{ij}(\tau,\tau'),
\end{aligned}
\tag{88}
$$

It is not immediately clear to what extent these additional non-canonical contributions permit a convenient reorganisation of correlations. We leave this as an interesting direction for further study.

We also comment that the consistency issue encountered here may be attributed to the use of the operator identity $\boldsymbol{n} = \boldsymbol{a}^\dagger\boldsymbol{a}$ in the evaluation of Eq. (85), whereas aside from this we have limited ourselves to the algebraic relations defining the DOF, Eq. (29). An alternative route forward is to use the exact equivalence of both sides of Eq. (84) with the dynamical susceptibility $\chi_{ij}(\tau,\tau')$. It is not straightforward to organise correlations for $\chi_{ij}(\tau,\tau')$ directly as the $\boldsymbol{n}$ may commute to zero, but it can instead be convenient to proceed via the Kubo–Mori relaxation function [61–63], or equivalently to analyse density correlations in combination with current correlations. Indeed this is employed extensively in the study of collective modes in the works of Plakida and coworkers [2]. It may prove instructive to reformulate this within the Schwinger approach, and we leave this as another interesting direction for further study.

## 5.4 Screening of density correlations

A great power of the canonical framework is its diagrammatic formulation, which allows for an intuitive analysis of the resummation of correlations. Such resummations can nevertheless be equivalently formulated within the Schwinger approach. In this section we set aside the question of how to generate conserving approximations, and now focus on making a resummation of density-induced

correlations [1, 64, 65] within our formalism. In doing so we obtain a screening approximation which can be viewed as an analogue of the commonly employed RPA and GW approximations for canonical DOFs.

We return to the equations of motion for $\mathcal{G}$ above, Eqs. (70). We now proceed by introducing effective sources, combining the density contributions with the external sources,

$$
\begin{aligned}
\xi_{ij}(\tau) &= \delta_{ij}\big(\zeta_i(\tau) - V_{il}\langle \boldsymbol{n_l}(\tau)\rangle\big) + \lambda\varepsilon_{ij}\langle \boldsymbol{n_i}(\tau)\rangle, \\
\overline{\xi}_{ij}(\tau) &= \delta_{ij}\big(\zeta_i(\tau) - \langle \boldsymbol{n_l}(\tau)\rangle V_{li}\big) + \lambda\varepsilon_{ij}\langle \boldsymbol{n_j}(\tau)\rangle.
\end{aligned}
\tag{89}
$$

It is then convenient to re-express Eqs. (71) as follows

$$
\begin{aligned}
\mathcal{D}_{ij}(\tau,\tau') &= \big(-\delta_{ij}\partial_\tau + \xi_{ij}(\tau) - \varepsilon_{ij}\big)\delta(\tau-\tau') - \mathcal{R}_{ij}(\tau,\tau'), \\
\overline{\mathcal{D}}_{ij}(\tau,\tau') &= \big(-\delta_{ij}\partial_\tau + \overline{\xi}_{ij}(\tau) - \varepsilon_{ij}\big)\delta(\tau-\tau') - \overline{\mathcal{R}}_{ij}(\tau,\tau').
\end{aligned}
\tag{90}
$$

with

$$
\begin{aligned}
\mathcal{R}_{ij}(\tau,\tau') &= \lambda\varepsilon_{il}\mathcal{G}_{ll'}(\tau,\tilde{\boldsymbol{\tau}})\nabla_i(\tau)\mathcal{G}^{-1}_{l'j}(\tilde{\boldsymbol{\tau}},\tau') - V_{il}\mathcal{G}_{il'}(\tau,\tilde{\boldsymbol{\tau}})\nabla_l(\tau)\mathcal{G}^{-1}_{l'j}(\tilde{\boldsymbol{\tau}},\tau'), \\
\overline{\mathcal{R}}_{ij}(\tau,\tau') &= \lambda\big(\nabla_j(\tau')\mathcal{G}^{-1}_{il'}(\tau,\tilde{\boldsymbol{\tau}})\big)\mathcal{G}_{l'l}(\tilde{\boldsymbol{\tau}},\tau')\varepsilon_{lj} - \big(\nabla_l(\tau')\mathcal{G}^{-1}_{il'}(\tau,\tilde{\boldsymbol{\tau}})\big)\mathcal{G}_{l'j}(\tilde{\boldsymbol{\tau}},\tau')V_{lj}.
\end{aligned}
\tag{91}
$$

We further introduce effective vertices

$$
\Gamma_{ij,kl}(\tau,\tau';\tau'') = \frac{\delta\mathcal{G}^{-1}_{ij}(\tau,\tau')}{\delta\xi_{kl}(\tau'')}, \qquad \overline{\Gamma}_{kl,ij}(\tau'';\tau,\tau') = \frac{\delta\mathcal{G}^{-1}_{ij}(\tau,\tau')}{\delta\overline{\xi}_{kl}(\tau'')},
\tag{92}
$$

related to the bare vertex as

$$
\nabla_l(\tau'')\mathcal{G}^{-1}_{ij}(\tau,\tau') = \Gamma_{ij,\boldsymbol{kk'}}(\tau,\tau';\tilde{\boldsymbol{\tau}})\mathcal{Y}_{\boldsymbol{kk'},l}(\tilde{\boldsymbol{\tau}},\tau'') = \overline{\mathcal{Y}}_{l,\boldsymbol{kk'}}(\tau'',\tilde{\boldsymbol{\tau}})\overline{\Gamma}_{\boldsymbol{kk'},ij}(\tilde{\boldsymbol{\tau}};\tau,\tau'),
\tag{93}
$$

where

$$
\mathcal{Y}_{ij,l}(\tau,\tau') = \nabla_l(\tau')\xi_{ij}(\tau), \qquad \overline{\mathcal{Y}}_{l,ij}(\tau,\tau') = \nabla_l(\tau)\overline{\xi}_{ij}(\tau').
\tag{94}
$$

Maintaining the logic of Sec. 5.2 above, Eq. (91) then takes the form

$$
\begin{aligned}
\mathcal{R}_{ij}(\tau,\tau') &= \lambda\varepsilon_{il}\mathcal{G}_{ll'}(\tau,\tilde{\boldsymbol{\tau}})\overline{\mathcal{Y}}_{i,\boldsymbol{kk'}}(\tau,\tilde{\boldsymbol{\tau}}')\overline{\Gamma}_{\boldsymbol{kk'},l'j}(\tilde{\boldsymbol{\tau}}';\tilde{\boldsymbol{\tau}},\tau') - V_{il}\mathcal{G}_{il'}(\tau,\tilde{\boldsymbol{\tau}})\overline{\mathcal{Y}}_{l,\boldsymbol{kk'}}(\tau,\tilde{\boldsymbol{\tau}}')\overline{\Gamma}_{\boldsymbol{kk'},l'j}(\tilde{\boldsymbol{\tau}}';\tilde{\boldsymbol{\tau}},\tau') \\
\overline{\mathcal{R}}_{ij}(\tau,\tau') &= \lambda\Gamma_{il',\boldsymbol{kk'}}(\tau,\tilde{\boldsymbol{\tau}};\tilde{\boldsymbol{\tau}}')\mathcal{Y}_{\boldsymbol{kk'},j}(\tilde{\boldsymbol{\tau}}',\tau')\mathcal{G}_{l'l}(\tilde{\boldsymbol{\tau}},\tau')\varepsilon_{lj} - \Gamma_{il',\boldsymbol{kk'}}(\tau,\tilde{\boldsymbol{\tau}};\tilde{\boldsymbol{\tau}}')\mathcal{Y}_{\boldsymbol{kk'},l}(\tilde{\boldsymbol{\tau}}',\tau')\mathcal{G}_{l'j}(\tilde{\boldsymbol{\tau}},\tau')V_{lj}.
\end{aligned}
\tag{95}
$$

The objects $\mathcal{Y}_{ij,l}(\tau,\tau')$ and $\overline{\mathcal{Y}}_{ij,l}(\tau,\tau')$ are analogues of the inverse dielectric function, and upon evaluating the variational derivative we can relate them to the dynamical susceptibility $\chi_{ij}(\tau,\tau') = \langle \boldsymbol{n_i}(\tau)\boldsymbol{n_j}(\tau')\rangle - \langle \boldsymbol{n_i}(\tau)\rangle\langle \boldsymbol{n_j}(\tau')\rangle = \nabla_i(\tau)\langle \boldsymbol{n_j}(\tau')\rangle = \nabla_j(\tau')\langle \boldsymbol{n_i}(\tau)\rangle$ as follows

$$
\begin{aligned}
\mathcal{Y}_{ij,l}(\tau,\tau') &= \delta_{ij}\delta_{il}\delta(\tau-\tau') - \delta_{ij}V_{i\boldsymbol{k}}\chi_{\boldsymbol{kl}}(\tau,\tau') + \lambda\varepsilon_{ij}\chi_{il}(\tau,\tau'), \\
\overline{\mathcal{Y}}_{l,ij}(\tau,\tau') &= \delta_{ij}\delta_{il}\delta(\tau-\tau') - \delta_{ij}V_{\boldsymbol{k}i}\chi_{l\boldsymbol{k}}(\tau,\tau') + \lambda\varepsilon_{ij}\chi_{lj}(\tau,\tau').
\end{aligned}
\tag{96}
$$

Let us further define analogues of the polarisation

$$\mathcal{P}_{l,ij}(\tau,\tau') = \frac{\delta\langle \boldsymbol{n}_l(\tau)\rangle}{\delta\xi_{ij}(\tau')}, \qquad \overline{\mathcal{P}}_{ij,l}(\tau,\tau') = \frac{\delta\langle \boldsymbol{n}_l(\tau')\rangle}{\delta\overline{\xi}_{ij}(\tau)}. \tag{97}$$

Then expressing the dynamical susceptibility through variations of the effective source

$$\chi_{ij}(\tau,\tau') = \mathcal{P}_{i,\boldsymbol{kk'}}(\tau,\tilde{\boldsymbol{\tau}})\mathcal{Y}_{\boldsymbol{kk'},j}(\tilde{\boldsymbol{\tau}},\tau') = \overline{\mathcal{Y}}_{i,\boldsymbol{kk'}}(\tau,\tilde{\boldsymbol{\tau}})\overline{\mathcal{P}}_{\boldsymbol{kk'},j}(\tilde{\boldsymbol{\tau}},\tau'), \tag{98}$$

leads to a pair of closed equations for $\chi$,

$$\begin{aligned}
\chi_{ij}(\tau,\tau') &= \mathcal{P}_{i,jj}(\tau,\tau') - \mathcal{P}_{i,\boldsymbol{kk}}(\tau,\tilde{\boldsymbol{\tau}})V_{\boldsymbol{kl}}\chi_{lj}(\tilde{\boldsymbol{\tau}},\tau') + \lambda\mathcal{P}_{i,\boldsymbol{kl}}(\tau,\tilde{\boldsymbol{\tau}})\varepsilon_{\boldsymbol{kl}}\chi_{\boldsymbol{kj}}(\tilde{\boldsymbol{\tau}},\tau'), \\
\chi_{ij}(\tau,\tau') &= \overline{\mathcal{P}}_{ii,j}(\tau,\tau') - \chi_{il}(\tau,\tilde{\boldsymbol{\tau}})V_{\boldsymbol{lk}}\overline{\mathcal{P}}_{\boldsymbol{kk},j}(\tilde{\boldsymbol{\tau}},\tau') + \lambda\chi_{il}(\tau,\tilde{\boldsymbol{\tau}})\varepsilon_{\boldsymbol{kl}}\overline{\mathcal{P}}_{\boldsymbol{kl},j}(\tilde{\boldsymbol{\tau}},\tau').
\end{aligned} \tag{99}$$

These equations capture the resummation of density-induced correlations. For a conserving approximation the results they yield are equivalent.

We now focus on the simplest approximation which is to take the leading contribution to the effective vertices

$$\begin{aligned}
\Gamma_{ij,kl}(\tau,\tau';\tau'') &= \mathcal{I}_{ik}^{-1}(\tau,\tau')\delta_{jl}\delta(\tau'-\tau''), \\
\overline{\Gamma}_{kl,ij}(\tau'';\tau,\tau') &= \delta_{ik}\delta(\tau-\tau'')\mathcal{I}_{lj}^{-1}(\tau,\tau').
\end{aligned} \tag{100}$$

We also now set the external sources $\zeta_i(\tau)$ to zero. It is convenient here to write $\mathcal{I}_{ij}(\tau,\tau') = \delta(\tau-\tau')\delta_{ij}\mathcal{I}$, with $\mathcal{I} = 1 - \lambda\langle \boldsymbol{n}\rangle$ which is now independent of both $i$ and $\tau$. Considering first the resummed dynamical susceptibility, we employ $\langle \boldsymbol{n}_i(\tau)\rangle = \langle \boldsymbol{a}_i^\dagger(\tau)\boldsymbol{a}_i(\tau)\rangle = \mp\mathcal{G}_{ii}(\tau,\tau^+)$ to write

$$\mathcal{P}_{l,ij}(\tau,\tau') = \pm\mathcal{G}_{li}(\tau,\tau')\mathcal{I}^{-1}\mathcal{G}_{jl}(\tau',\tau), \qquad \overline{\mathcal{P}}_{ij,l}(\tau,\tau') = \pm\mathcal{G}_{li}(\tau',\tau)\mathcal{I}^{-1}\mathcal{G}_{jl}(\tau,\tau'). \tag{101}$$

The pair of closed Eqs. (99) then become

$$\begin{aligned}
\chi_{ij}(\tau,\tau') &= \pm\mathcal{G}_{ij}(\tau,\tau')\mathcal{I}^{-1}\mathcal{G}_{ji}(\tau',\tau) \mp \mathcal{G}_{i\boldsymbol{k}}(\tau,\tilde{\boldsymbol{\tau}})\mathcal{I}^{-1}\mathcal{G}_{\boldsymbol{k}i}(\tilde{\boldsymbol{\tau}},\tau)V_{\boldsymbol{kl}}\chi_{lj}(\tilde{\boldsymbol{\tau}},\tau') \\
&\quad \pm \lambda\mathcal{G}_{i\boldsymbol{k}}(\tau,\tilde{\boldsymbol{\tau}})\mathcal{I}^{-1}\varepsilon_{\boldsymbol{kl}}\mathcal{G}_{li}(\tilde{\boldsymbol{\tau}},\tau)\chi_{\boldsymbol{kj}}(\tilde{\boldsymbol{\tau}},\tau'), \\
\chi_{ij}(\tau,\tau') &= \pm\mathcal{G}_{ji}(\tau',\tau)\mathcal{I}^{-1}\mathcal{G}_{ij}(\tau,\tau') \mp \chi_{il}(\tau,\tilde{\boldsymbol{\tau}})V_{\boldsymbol{lk}}\mathcal{G}_{j\boldsymbol{k}}(\tau',\tilde{\boldsymbol{\tau}})\mathcal{I}^{-1}\mathcal{G}_{\boldsymbol{k}j}(\tilde{\boldsymbol{\tau}},\tau') \\
&\quad \pm \lambda\chi_{il}(\tau,\tilde{\boldsymbol{\tau}})\mathcal{G}_{j\boldsymbol{k}}(\tau',\tilde{\boldsymbol{\tau}})\varepsilon_{\boldsymbol{kl}}\mathcal{I}^{-1}\mathcal{G}_{lj}(\tilde{\boldsymbol{\tau}},\tau'),
\end{aligned} \tag{102}$$

which upon Fourier transforming give

$$\chi_p(\mathrm{i}\nu_n) = \frac{\mathcal{X}_p(\mathrm{i}\nu_n)}{1 + V_p\mathcal{X}_p(\mathrm{i}\nu_n) - \lambda\mathcal{X}_p^\varepsilon(\mathrm{i}\nu_n)}, \tag{103}$$

where

$$\mathcal{X}_p(\mathrm{i}\nu_n) = \pm \tfrac{1}{\beta\mathcal{V}} \sum_{m,q} \mathcal{G}_{p+q}(\mathrm{i}\nu_n + \mathrm{i}\omega_m)\mathcal{I}^{-1}\mathcal{G}_q(\mathrm{i}\omega_m),$$

$$\mathcal{X}_p^\varepsilon(\mathrm{i}\nu_n) = \pm \tfrac{1}{\beta\mathcal{V}} \sum_{m,q} \mathcal{G}_{p+q}(\mathrm{i}\nu_n + \mathrm{i}\omega_m)\mathcal{I}^{-1}\varepsilon_q\mathcal{G}_q(\mathrm{i}\omega_m). \tag{104}$$

This generalises the canonical RPA approximation (which is reobtained at $\lambda = 0$) to non-canonical DOFs.

Finally, we return to $\mathcal{R}$ and $\overline{\mathcal{R}}$ of Eqs. (95). Substituting the approximation of Eqs. (100), and Fourier transforming, they both lead to the same approximation, conveniently expressed as

$$\mathcal{K}_p^\star = \varepsilon_p\mathcal{I}^2 + V_0\mathcal{I}\langle\boldsymbol{n}\rangle - \tfrac{1}{\beta\mathcal{V}}\sum_{m,q}(V_q - \lambda\varepsilon_{p-q})\mathcal{G}_{p-q}(\mathrm{i}\omega_m),$$

$$\mathcal{M}_p^\star(\mathrm{i}\omega_n) = \tfrac{1}{\beta\mathcal{V}}\sum_{m,q}(V_q - \lambda\varepsilon_{p-q})^2\chi_q(\mathrm{i}\nu_m)\mathcal{G}_{p-q}(\mathrm{i}\omega_n - \mathrm{i}\nu_m). \tag{105}$$

The expression for $\mathcal{K}^\star$ suppresses the subleading static susceptibility contributions from Eq. (57). The expression for $\mathcal{M}^\star$ gives the mode-coupling approximation referred to above in our discussion of Eq. (58), with the improvement that here $\chi$ takes the RPA form of Eq. (103). This approximation can be viewed as a non-canonical analogue of the GW approximation [64,65], which is reobtained at $\lambda = 0$ as follows

$$\mathcal{G}_p(\mathrm{i}\omega_n) = \frac{1}{\mathrm{i}\omega_n + \mu - \varepsilon_p - V_0\langle\boldsymbol{n}\rangle + \tfrac{1}{\beta\mathcal{V}}\sum_{m,q}\mathcal{G}_{p-q}(\mathrm{i}\omega_n - \mathrm{i}\nu_m)\mathcal{W}_q(\mathrm{i}\nu_m)}, \tag{106}$$

where $V_0\langle\boldsymbol{n}\rangle$ is the Hartree contribution and $\mathcal{W}_p(\mathrm{i}\nu_n) = V_p - V_p^2\chi_p(\mathrm{i}\nu_n) = \frac{V_p}{1+V_p\mathcal{X}_p(\mathrm{i}\nu_n)}$ is the dynamically screened exchange interaction. For a non-canonical DOF the approximation does not admit such a simple interpretation as a screening of the interaction, but it is not so much more complicated either. We leave detailed study of specific applications of this approximation to future work.

# 6  Discussion

We stop short of investigating specific applications. Indeed there already exist extensive bodies of work demonstrating the usefulness of non-canonical DOFs for magnetically ordered systems [6–10], as well as for the pseudogap regime of the cuprates [2,3]. Our purpose here is not to redo these works, but rather to more generally justify their approach so that a broader consensus on their value can be formed.

In this Discussion we wish instead to explore in general terms what the non-canonical paradigm can offer. To proceed let us adopt the perspective of assuming that non-canonical algebras do indeed provide legitimate quantum DOFs, to see where this leads. We will find that the interplay between canonical and non-canonical regimes provides a powerful principle for organising the phase diagram of correlated electronic behaviour, as summarised in Fig. 1.

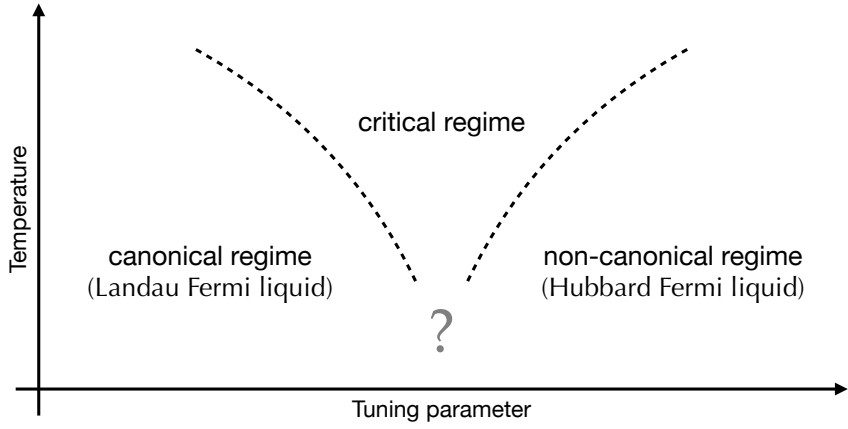

Figure 1: A schematic phase diagram for the metallic behaviour of correlated electrons. There are two distinct ways of organising electronic correlations, the canonical (Landau) description and the non-canonical (Hubbard) description, and critical behaviour emerges where these compete. At low temperature there may be a first-order transition, as seen for example within DMFT [66], or more generally this may be hidden by an ordered phase, for example $d$-wave superconductivity in the case of the cuprate compounds [67].

Canonical and non-canonical DOFs offer different ways of organising correlations, i.e. they lead to distinct quasi-particle descriptions. This is most clearly the case in the electronic and local moment settings, where the electron's DOF can be cast in either a canonical or non-canonical form as discussed in Secs. 2.2 and 2.3. To better discuss the interplay between the two, it is useful to first consider some limits to their range of applicability.

We can associate the breakdown of a quasi-particle description with the emergence of critical correlations in a system. Perhaps the most prominent instance of this is the occurrence of second order phase transitions. Here universal power-law correlations take over, as most conveniently characterised through the formalism of renormalisation group [68]. It is illustrative to take briefly the example of spin systems. As discussed throughout this paper, the spin-wave quasi-particle description of magnetically ordered systems may be formulated through the non-canonical spin DOF of Eq. (6). As the transition temperature is approached however, we have that $\langle \boldsymbol{n} \rangle$ approaches $S$, leading to the breakdown of the perturbative expansion underlying the quasi-particle description concurrent with the divergence of the correlation length and the emergence of critical correlations.

In a similar vein, a quasi-particle description generically breaks down as a system's geometry becomes strictly 1D [69–71]. Here again criticality emerges, with low-energy gapless behaviour characterised by the conformally invariant Luttinger liquid. The underlying DOFs are fractionalised, as is clearly seen for certain high-symmetry models solvable by Bethe ansatz, which correspond to limits where the excited modes scatter elastically [72,73]. In spin systems magnons deconfine into spinons [74,75], while electronic systems exhibit spin-charge separation [76,77]. Indeed, in passing let us remark more generally that we may not expect local DOFs as discussed in this paper to be appropriate for phenomena characterised by fractionalisation, e.g. quantum spin liquids [78] and fractional quantum Hall states [79].

Another key example is the Kondo effect. The critical nature of this phenomena has been

established through Wilson's renormalisation group analysis [80], the Bethe ansatz solution [81,82], and the conformal field theory description [83,84]. These place it beyond reach of a quasi-particle description. Nevertheless, Nozières demonstrated that the low-energy fixed-point admits a Fermi liquid description where one electron forms a singlet with the impurity spin [85] (c.f. Eq. (130) of App. A). On the other hand, the high-energy fixed point corresponds to a canonical Fermi liquid and a decoupled spin. The emergence of criticality at the Kondo temperature arises from the competition between these two distinct ways of organising correlations.

A closely related phenomena is the Mott metal-insulator transition [86,87]. This occurs when electronic correlations induce the opening of a gap within an electronic band, in conflict with the essence of Landau's conception of an electronic quasi-particle. A powerful framework for addressing this is dynamical mean-field theory (DMFT) [66]. The method is exact in infinite dimensions, where spatial correlations are suppressed but dynamical correlations survive. DMFT allows for the computation of the local Green's function through a self-consistent mapping to an Anderson impurity model [88]. This mapping is formal, it does not reflect any choice of underlying DOFs. Indeed the Anderson impurity model exhibits the Kondo effect, and in general it is solved via unbiased numerical methods. DMFT thus offers an approach which is complementary to a quasi-particle description. Although often its results are interpreted through a quasi-particle lens, its great strength lies in its ability to characterise phenomena beyond the reach of a quasi-particle framework. Its limitation however is that, being inherently local, it is not well-suited to capturing spatial correlations, and for this task a quasi-particle description is more appropriate. Let us proceed to briefly summarise what DMFT tells us, and then discuss how this can be interpreted through the interplay of the canonical and non-canonical quasi-particle descriptions.

Within the context of the Hubbard model, single-site DMFT exhibits a Mott transition between metallic and insulating states which is first-order at low temperatures [66]. Critical scaling is observed in the crossover region at temperatures above the end-point of the first-order transition [89–91]. More modern cluster DMFT studies indicate that the low-temperature first-order transition extends away from the Mott transition to finite doping [92,93]. Here the Mott gap is seen to border an unconventional metallic state, with the first-order transition to a conventional metallic state occurring subsequently upon increased doping. The first-order transition may also get hidden behind an ordered phase [94,95].

These results admit a natural interpretation in view of the two formulations of the electronic DOF discussed in Sec. 2.2. The conventional metallic state is straightforwardly identified as the canonical quasi-particle regime, i.e. as a Landau Fermi liquid. It is similarly automatic to identify the unconventional metallic state as that of non-canonical electronic quasi-particles, i.e. as a Hubbard Fermi liquid. Indeed we recall that this non-canonical regime manifests a splitting of the electron, Eq. (18), which inherently describes a metallic state bordering a Mott gap. We thus interpret the DMFT results as indicating a first-order transition between the canonical (Landau) and non-canonical (Hubbard) quasi-particle regimes at low temperature (which may get hidden behind an ordered phase), along with a critical regime at higher temperatures.

We attribute the critical regime to competition between the two distinct ways of organising electronic correlations, as summarised in Fig. 1. This is similar in spirit to the notions of local

quantum criticality [96] and Mott quantum criticality [90], and is also reminiscent of the holographic description of quantum criticality [58, 97]. It may be convenient to visualise its emergence as follows. Distinct quasi-particle descriptions represent distinct structures in the energy spectrum. To continuously interpolate between two quasi-particle regimes necessitates that these structures get washed out in between, giving rise to a maximally ergodic regime (up to global symmetries) exhibiting critical dynamical correlations. In the absence of a quasi-particle structure, such a regime could be characterised by a hydrodynamic description accounting for the diffusion of the globally conserved charges, see e.g. [58].

We have focused the discussion on the theoretical description of electronic systems. Let us round out by highlighting that the above analysis is mirrored by the experimental situation. A unifying theme in the study of strongly correlated materials is the observation of distinct metallic regimes separated by a regime of quantum criticality. Hallmark examples are the cuprates, where a critical 'strange metal' lies between the pseudogap and Fermi liquid metallic regimes [67], and the heavy-fermion compounds where a similarly quantum critical regime is seen to separate conventional Fermi liquid and heavy Fermi liquid regimes [98]. In both cases, a low-temperature transition between the distinct metallic regimes is unambiguously observed in Hall coefficient measurements [99, 100]. These systems are thus naturally interpreted through the generic phase diagram of Fig. 1, with the pseudogap and heavy Fermi liquid regimes identified as potential examples of Hubbard Fermi liquids. This is indeed consistent with the theoretical studies cited above [2–4].

By contrast, the more conventional way to characterise the phase diagrams of these systems is to attribute the critical regime to an underlying quantum critical point [101–105], i.e. to a continuous phase transition taking place at zero temperature. This has some relevance for the heavy-fermion compounds where there may occur a magnetic transition concurrent with the transition between conventional and heavy Fermi liquid regimes [104]. It has perhaps clouded interpretations of systems such as the cuprates however, where all evidence points against the existence of an analogous quantum critical point, even one of a hidden variety [99]. If though we instead attribute the critical behaviour to competition between quasi-particle regimes we reconcile this discrepancy in a simple manner. A magnetic transition may occur at the quasi-particle transition in the heavy-fermion case as in the canonical regime there is a spin-moment which is free to order, whereas this gets entwined into the electronic DOF in the non-canonical regime. There is no analogue in an effective purely electronic setting as neither formulation of the electronic DOF leave something independent of it.

# 7    Conclusion

In this paper we have addressed interacting quantum systems from the perspective of how correlations may be organised. We have focused in particular on whether non-canonical algebras can provide legitimate quantum DOFs. While we have not obtained a definitive answer, we have seen that this is still very much a work in progress. We have obtained a closed expression, Eqs. (75)-(82), for a self-energy-like object $M_p^\star(\omega)$ appearing in the Dyson form of Eq. (58). We further made an RPA-like resummation of density-induced correlations, Eq. (103), leading to a GW-like approximation for the Green's function, Eq. (105), which has a comparable level of complexity to

its canonical counterpart. We have also highlighted issues related to generating conserving approximations, and have discussed routes forward related to the study of vertex corrections, collective modes and transport.

We have provided a coherent description of key examples of non-canonical DOFs, those for spin, electron and local moment systems. We have emphasised the appearance of parameters $\lambda$ inherent to their respective algebras, and have discussed the role these play in organising correlations. We have also highlighted large-$S$ limits suitable for characterising spin and charge order of electronic origin, which have yet to be analysed.

We have paid particular attention to the non-canonical formulation of the electronic DOF. We have argued in favour of a non-canonical Hubbard Fermi liquid, distinct from the canonical Landau Fermi liquid. We have clarified how this provides a unified description of electron and local moment systems. We have complemented our analysis with discussions of both the emergence of criticality and of DMFT, resulting in a proposal of a generic phase diagram for the metallic behaviour of correlated electrons, Fig. 1.

**Acknowledgements.** We are grateful to M. Grandadam and C. Pépin for useful discussions, and E. Ilievski and N. Plakida for valuable comments on the manuscript. This work is supported by the ANR IDTODQG project grant ANR-16-CE91-0009 of the French Agence Nationale de la Recherche.

# A    Lie superalgebra $u(N|M)$

In Sec. 2 we described non-canonical DOFs for spin, electron, and local moment systems. In this appendix we provide a more formal overview of the underlying algebraic structures, along with their representations. This allows for a better understanding of the origins of the non-canonical DOFs, and also reveals how they relate to certain slave-particle formulations of lattice models frequently invoked in theoretical studies of condensed matter systems [106–108].

We focus on the family of Lie superalgebras $u(N|M)$ (we adopt a physicist's convention and do not distinguish between $u(N|M)$ and $gl(N|M)$, nor between $su(N|M)$ and $sl(N|M)$). Firstly, the algebra $su(N|M)$ is generated by a set of operators $\boldsymbol{Q}_\alpha^a$, $\boldsymbol{Q}^{\dagger\alpha}_a$, $\boldsymbol{L}_b^a$, $\boldsymbol{R}_\beta^\alpha$ and $\boldsymbol{C}$, with $a = 1, 2, \ldots, N$ and $\alpha = 1, 2, \ldots, M$. The pair $\boldsymbol{Q}_\alpha^a$ and $\boldsymbol{Q}^{\dagger\alpha}_a$ are fermionic, obeying the anti-commutation relations

$$\{\boldsymbol{Q}_\alpha^a, \boldsymbol{Q}^{\dagger\beta}_b\} = \delta_b^a \delta_\alpha^\beta \boldsymbol{C} + \delta_\alpha^\beta \boldsymbol{L}_b^a + \delta_b^a \boldsymbol{R}_\alpha^\beta. \tag{107}$$

The $\boldsymbol{L}_b^a$ and $\boldsymbol{R}_\beta^\alpha$ generate two bosonic sub-algebras, $su(N)$ and $su(M)$ respectively,

$$
\begin{aligned}
&[\boldsymbol{L}_b^a, \boldsymbol{J}^c] = \delta_b^c \boldsymbol{J}^a - \tfrac{1}{N}\delta_b^a \boldsymbol{J}^c, &\quad& [\boldsymbol{R}_\beta^\alpha, \boldsymbol{J}^\gamma] = \delta_\beta^\gamma \boldsymbol{J}^\alpha - \tfrac{1}{M}\delta_\beta^\alpha \boldsymbol{J}^\gamma, \\
&[\boldsymbol{L}_b^a, \boldsymbol{J}_c] = -\delta_c^a \boldsymbol{J}_b + \tfrac{1}{N}\delta_b^a \boldsymbol{J}^c, &\quad& [\boldsymbol{R}_\beta^\alpha, \boldsymbol{J}_\gamma] = -\delta_\gamma^\alpha \boldsymbol{J}^\beta + \tfrac{1}{M}\delta_\beta^\alpha \boldsymbol{J}_\gamma,
\end{aligned}
\tag{108}
$$

where $\boldsymbol{J}$ denotes any generator with appropriate index. The generator $\boldsymbol{C}$ obeys

$$[\boldsymbol{C}, \boldsymbol{Q}_\alpha^a] = \tfrac{M-N}{NM}\boldsymbol{Q}_\alpha^a, \qquad [\boldsymbol{C}, \boldsymbol{Q}^{\dagger\alpha}_a] = \tfrac{N-M}{NM}\boldsymbol{Q}^{\dagger\alpha}_a, \tag{109}$$

and is central if $N = M$, i.e. it then commutes with all generators. The algebra $su(N|M)$ is extended to $u(N|M)$ by incorporating an additional generator $\boldsymbol{D}$ obeying

$$[\boldsymbol{D}, \boldsymbol{Q}^{\dagger\alpha}{}_a] = \boldsymbol{Q}^{\dagger\alpha}{}_a, \qquad [\boldsymbol{D}, \boldsymbol{Q}^a_\alpha] = -\boldsymbol{Q}^a_\alpha, \tag{110}$$

and commuting with all other generators.

It is useful to introduce an oscillator (slave-particle) realisation of $u(N|M)$ as follows

$$\begin{aligned}
\boldsymbol{Q}^a_\alpha &= \boldsymbol{b}^\dagger_a \boldsymbol{f}_\alpha, \quad \boldsymbol{Q}^{\dagger\alpha}{}_a = \boldsymbol{f}^\dagger_\alpha \boldsymbol{b}_a, \\
\boldsymbol{L}^a_b &= \boldsymbol{b}^\dagger_a \boldsymbol{b}_b - \tfrac{1}{N}\delta^a_b \boldsymbol{b}^\dagger_c \boldsymbol{b}_c, \quad \boldsymbol{R}^\alpha_\beta = \boldsymbol{f}^\dagger_\alpha \boldsymbol{f}_\beta - \tfrac{1}{M}\delta^\alpha_\beta \boldsymbol{f}^\dagger_\gamma \boldsymbol{f}_\gamma, \\
\boldsymbol{C} &= \tfrac{1}{N}\boldsymbol{b}^\dagger_a \boldsymbol{b}_a + \tfrac{1}{M}\boldsymbol{f}^\dagger_\alpha \boldsymbol{f}_\alpha, \\
\boldsymbol{D} &= -\tfrac{1}{2}\boldsymbol{b}^\dagger_a \boldsymbol{b}_a + \tfrac{1}{2}\boldsymbol{f}^\dagger_\alpha \boldsymbol{f}_\alpha,
\end{aligned} \tag{111}$$

where $\boldsymbol{b}_a$ and $\boldsymbol{f}_\alpha$ are canonical bosons and fermions respectively, summation over repeated indices is implied, and $\boldsymbol{D}$ is defined up to a constant shift. In the following we shall denote the common vacuum of $\boldsymbol{b}^\dagger_a$ and $\boldsymbol{f}^\dagger_\alpha$ by $|\Omega\rangle$. As $\boldsymbol{b}^\dagger_a \boldsymbol{b}_a + \boldsymbol{f}^\dagger_\alpha \boldsymbol{f}_\alpha$ commutes with all generators, a representation can be constructed for each positive integer $\mathcal{N}$ by restricting to the space of states obeying the constraint

$$\boldsymbol{b}^\dagger_a \boldsymbol{b}_a + \boldsymbol{f}^\dagger_\alpha \boldsymbol{f}_\alpha = \mathcal{N}. \tag{112}$$

Representations obtained in this way are commonly referred to as 'atypical' or 'short'. In particular, if $N = M$ then $\boldsymbol{C}$ is central and its eigenvalue is fixed through $NC = \mathcal{N}$, referred to as the shortening condition.

The fundamental representation of $u(N|M)$ is $(N+M)$-dimensional. In the oscillator realisation this corresponds to taking $\mathcal{N} = 1$, i.e. the basis is given by the one-particle states. It is instructive also to consider a matrix realisation for the fundamental representation. The generators of $su(N|M)$ are then regarded as the $(N + M) \times (N + M)$ matrices with zero supertrace, where for a general matrix $\boldsymbol{M} = \left(\begin{array}{c|c} \mathcal{A} & \mathcal{B} \\ \hline \mathcal{C} & \mathcal{D} \end{array}\right)$ the condition of zero supertrace is $\operatorname{str} \boldsymbol{M} = \operatorname{tr} \mathcal{A} - \operatorname{tr} \mathcal{D} = 0$. Schematically the generators take the form

$$\boldsymbol{Q} = \left(\begin{array}{c|c} 0 & * \\ \hline 0 & 0 \end{array}\right), \quad \boldsymbol{Q}^\dagger = \left(\begin{array}{c|c} 0 & 0 \\ \hline * & 0 \end{array}\right), \quad \boldsymbol{L} = \left(\begin{array}{c|c} * & 0 \\ \hline 0 & 0 \end{array}\right), \quad \boldsymbol{R} = \left(\begin{array}{c|c} 0 & 0 \\ \hline 0 & * \end{array}\right), \tag{113}$$

where $*$ denotes the existence of non-zero entries. The generator $\boldsymbol{C}$ is diagonal, and for $N = M$ it is proportional to the identity. The extension to $u(N|M)$ is given by

$$\boldsymbol{D} = \left(\begin{array}{c|c} -\tfrac{1}{2}\mathbb{I}_{N\times N} & 0 \\ \hline 0 & \tfrac{1}{2}\mathbb{I}_{M\times M} \end{array}\right), \tag{114}$$

which has non-zero supertrace.

Next we consider specific examples, which are both of interest in their own right and moreover serve to illustrate the general structure.

**Spin** $su(2)$: We can regard $su(2)$ in terms of $su(N|M)$ as either $N = 2$, $M = 0$ or $N = 0$, $M = 2$. In either case the fermionic generators $\boldsymbol{Q}$, $\boldsymbol{Q}^\dagger$ drop out, and the algebra reduces to one or other of the bosonic subalgebras.

In the first case $N = 2$, $M = 0$ the surviving non-trivial generators are $\boldsymbol{L}$. These can be related to $\vec{\boldsymbol{S}}$ through $\boldsymbol{S}^z = \boldsymbol{L}_2^2 = -\boldsymbol{L}_1^1$, $\boldsymbol{S}^+ = \boldsymbol{L}_1^2$, $\boldsymbol{S}^- = \boldsymbol{L}_2^1$. The oscillator realisation of $\boldsymbol{L}$ gives the Schwinger boson formulation of $su(2)$. The family of representations determined through the constraint $\boldsymbol{b}_1^\dagger \boldsymbol{b}_1 + \boldsymbol{b}_2^\dagger \boldsymbol{b}_2 = \mathcal{N}$, provide all $(2S + 1)$-dimensional multiplets $|S, m\rangle$ of $su(2)$, where $\mathcal{N} = 2S$ and $S$ is the magnitude of the spin. For each $\mathcal{N}$, the basis can be expressed explicitly as

$$|S, m\rangle = \frac{(\boldsymbol{b}_2^\dagger)^{S+m}(\boldsymbol{b}_1^\dagger)^{S-m}}{\sqrt{(S + m)!(S - m)!}} |\Omega\rangle, \qquad m \in \{-S, -S + 1, \ldots, S\}. \tag{115}$$

For the alternative case $N = 0$, $M = 2$ the $\boldsymbol{R}$ are the remaining generators. These are related to $\boldsymbol{S}$ similarly as above $\boldsymbol{S}^z = \boldsymbol{R}_2^2 = -\boldsymbol{R}_1^1$, $\boldsymbol{S}^+ = \boldsymbol{R}_1^2$, $\boldsymbol{S}^- = \boldsymbol{R}_2^1$. Here the oscillator realisation gives the Abrikosov fermion formulation for the spin-$1/2$ representation of $su(2)$. Due to the Pauli exclusion principle for fermions, the oscillator realisation here provides a non-trivial representation only for $\mathcal{N} = 1$, with basis given by the doublet

$$|\downarrow\rangle = \boldsymbol{f}_1^\dagger |\Omega\rangle, \quad |\uparrow\rangle = \boldsymbol{f}_2^\dagger |\Omega\rangle. \tag{116}$$

The matrix realisation of the generators for the fundamental representation of $su(2)$ over this basis are given by $\boldsymbol{S}^z = \begin{pmatrix} -\frac{1}{2} & 0 \\ 0 & \frac{1}{2} \end{pmatrix}$, $\boldsymbol{S}^+ = \begin{pmatrix} 0 & 0 \\ 1 & 0 \end{pmatrix}$, $\boldsymbol{S}^- = \begin{pmatrix} 0 & 1 \\ 0 & 0 \end{pmatrix}$. The extension to $u(2)$ is obtained by incorporating the identity, which commutes with all other generators.

**Canonical fermion** $u(1|1)$: The simplest Lie superalgebra is $su(1|1)$, which is none other than the familiar anti-commutation relation of canonical fermions, $\{\boldsymbol{c}, \boldsymbol{c}^\dagger\} = 1$. To see this we note that for $N = M = 1$ the two bosonic subalgebras trivialise, i.e. $\boldsymbol{L} = \boldsymbol{R} = 0$, and the only non-trivial relation is $\{\boldsymbol{Q}_1^1, \boldsymbol{Q}^{\dagger 1}_1\} = \boldsymbol{C}$, where $\boldsymbol{C} = C\mathbf{1}$ is proportional to the identity $\mathbf{1}$. The operator $\boldsymbol{D}$ extending the algebra to $u(1|1)$ further obeys $[\boldsymbol{D}, \boldsymbol{Q}^{\dagger 1}_1] = \boldsymbol{Q}^{\dagger 1}_1$ and $[\boldsymbol{D}, \boldsymbol{Q}_1^1] = -\boldsymbol{Q}_1^1$. We can thus interpret $\boldsymbol{c} = \frac{1}{\sqrt{C}}\boldsymbol{Q}_1^1$, $\boldsymbol{c}^\dagger = \frac{1}{\sqrt{C}}\boldsymbol{Q}^{\dagger 1}_1$ and $\boldsymbol{n}$ as $\boldsymbol{D}$, and in doing so we reobtain the canonical fermionic relations of Eq. (1). We remark that the representations obtained through the oscillator realisation are two dimensional for all $\mathcal{N}$, with basis

$$\frac{(\boldsymbol{b}_1^\dagger)^{\mathcal{N}}}{\sqrt{\mathcal{N}!}} |\Omega\rangle, \quad \frac{(\boldsymbol{b}_1^\dagger)^{\mathcal{N}-1}}{\sqrt{(\mathcal{N}-1)!}} \boldsymbol{f}_1^\dagger |\Omega\rangle, \tag{117}$$

and $C = \mathcal{N}$. For the $\mathcal{N} = 1$ representation we can identify $|0\rangle = \boldsymbol{b}_1^\dagger |\Omega\rangle$ and $\boldsymbol{c}^\dagger |0\rangle = \boldsymbol{f}_1^\dagger |\Omega\rangle$, and the corresponding matrix realisation is given by

$$\mathbf{1} = \begin{pmatrix} 1 & 0 \\ 0 & 1 \end{pmatrix}, \quad \boldsymbol{c} = \begin{pmatrix} 0 & 1 \\ 0 & 0 \end{pmatrix}, \quad \boldsymbol{c}^\dagger = \begin{pmatrix} 0 & 0 \\ 1 & 0 \end{pmatrix}, \quad \boldsymbol{n} = \begin{pmatrix} 0 & 0 \\ 0 & 1 \end{pmatrix}. \tag{118}$$

Comparing $u(1|1)$ with $su(2)$ we see that their $2 \times 2$ matrix realisations are essentially identical.

Let us emphasise for clarity then that their algebras are distinguished by their grading. This detail is not very significant when dealing with operators at the same site, but it becomes crucial when operators at different sites are involved. Bosonic generators at different sites commute to zero, whereas fermionic generators at different sites anti-commute to zero, with the consequence that $u(1|1)$ and $su(2)$ encode non-local correlations in distinct ways.

**Non-canonical electron** $u(2|2)$: The final example we take is $u(2|2)$ [47, 48, 109], which plays a central role in our discussion of electronic and local moment DOFs in Secs. 2.2 and 2.3. We begin by highlighting an important subtlety, which is that $u(2|2)$ admits an exceptional central extension. Whereas for general $u(N|M)$ the fermionic generators obey

$$\{\boldsymbol{Q}_\alpha^a, \boldsymbol{Q}_\beta^b\} = 0, \qquad \{\boldsymbol{Q}^{\dagger\alpha}_a, \boldsymbol{Q}^{\dagger\beta}_b\} = 0, \tag{119}$$

for the case of $u(2|2)$ these relations can be made non-trivial

$$\{\boldsymbol{Q}_\alpha^a, \boldsymbol{Q}_\beta^b\} = \epsilon_{\alpha\beta}\epsilon^{ab}\boldsymbol{A}, \qquad \{\boldsymbol{Q}^{\dagger\alpha}_a, \boldsymbol{Q}^{\dagger\beta}_b\} = \epsilon_{ab}\epsilon^{\alpha\beta}\boldsymbol{B}, \tag{120}$$

where the generators $\boldsymbol{A}$, $\boldsymbol{B}$ are central, and $\epsilon_{12} = -\epsilon_{21} = 1$, $\epsilon_{11} = \epsilon_{22} = 0$. This is special to $N = M = 2$, as only then are the antisymmetric tensors $\epsilon_{ab}$ and $\epsilon_{\alpha\beta}$ well-defined. So as not to labour notations we refer to this exceptionally extended algebra also as $u(2|2)$.

The extension deforms the oscillator realisation of the generators to

$$\begin{aligned}
\boldsymbol{Q}_\alpha^a &= u\,\boldsymbol{b}_a^\dagger \boldsymbol{f}_\alpha + v\,\epsilon^{ab}\epsilon_{\alpha\beta}\boldsymbol{f}_\beta^\dagger \boldsymbol{b}_b, & \boldsymbol{Q}^{\dagger\alpha}_a &= u^*\,\boldsymbol{f}_\alpha^\dagger \boldsymbol{b}_a + v^*\,\epsilon^{\alpha\beta}\epsilon_{ab}\boldsymbol{b}_b^\dagger \boldsymbol{f}_\beta, \\
\boldsymbol{L}_b^a &= \boldsymbol{b}_a^\dagger \boldsymbol{b}_b - \tfrac{1}{2}\delta_b^a \boldsymbol{b}_c^\dagger \boldsymbol{b}_c, & \boldsymbol{R}_\beta^\alpha &= \boldsymbol{f}_\alpha^\dagger \boldsymbol{f}_\beta - \tfrac{1}{2}\delta_\beta^\alpha \boldsymbol{f}_\gamma^\dagger \boldsymbol{f}_\gamma, \\
\boldsymbol{A} &= uv\,(\boldsymbol{b}_a^\dagger \boldsymbol{b}_a + \boldsymbol{f}_\alpha^\dagger \boldsymbol{f}_\alpha), & \boldsymbol{B} &= u^*v^*\,(\boldsymbol{b}_a^\dagger \boldsymbol{b}_a + \boldsymbol{f}_\alpha^\dagger \boldsymbol{f}_\alpha), \\
\boldsymbol{C} &= \tfrac{1}{2}(|u|^2 + |v|^2)(\boldsymbol{b}_a^\dagger \boldsymbol{b}_a + \boldsymbol{f}_\alpha^\dagger \boldsymbol{f}_\alpha), & & \\
\boldsymbol{D} &= -\tfrac{1}{2}\boldsymbol{b}_a^\dagger \boldsymbol{b}_a + \tfrac{1}{2}\boldsymbol{f}_\alpha^\dagger \boldsymbol{f}_\alpha, & &
\end{aligned} \tag{121}$$

where the deformation parameters are constrained to obey $|u|^2 - |v|^2 = 1$. We we can thus again construct representations for each positive integer $\mathcal{N}$ by restricting to the space of states obeying the constraint Eq. (112). Here the shortening condition becomes $2\sqrt{C^2 - AB} = \mathcal{N}$, where $A, B, C$ are the eigenvalues of $\boldsymbol{A}, \boldsymbol{B}, \boldsymbol{C}$.

The fundamental 4-dimensional representation is obtained by restricting to the single-particle states

$$\boldsymbol{b}_1^\dagger |\Omega\rangle, \quad \boldsymbol{b}_2^\dagger |\Omega\rangle, \quad \boldsymbol{f}_1^\dagger |\Omega\rangle, \quad \boldsymbol{f}_2^\dagger |\Omega\rangle. \tag{122}$$

The bosonic and fermionic states are respectively $su(2)$ doublets of $\boldsymbol{L}$ and $\boldsymbol{R}$. These are naturally identified with the four basis states of an electron

$$|\downarrow\rangle, \quad |\uparrow\rangle, \quad |\circ\rangle, \quad |\bullet\rangle, \tag{123}$$

from Eq. (9). This electronic interpretation of $u(2|2)$ is discussed in detail in Sec. 2.2. The fermionic

generators can be identified through

$$
\begin{aligned}
\boldsymbol{q}_{\uparrow\circ} &= \sqrt{\kappa}\,\boldsymbol{Q}_1^1, & \boldsymbol{q}_{\uparrow\bullet} &= -\sqrt{\kappa}\,\boldsymbol{Q}_2^1, & \boldsymbol{q}_{\uparrow\circ}^\dagger &= \sqrt{\kappa}\,\boldsymbol{Q}_1^{\dagger 1}, & \boldsymbol{q}_{\uparrow\bullet}^\dagger &= -\sqrt{\kappa}\,\boldsymbol{Q}_1^{\dagger 2}, \\
\boldsymbol{q}_{\downarrow\circ} &= \sqrt{\kappa}\,\boldsymbol{Q}_1^2, & \boldsymbol{q}_{\downarrow\bullet} &= -\sqrt{\kappa}\,\boldsymbol{Q}_2^2, & \boldsymbol{q}_{\downarrow\circ}^\dagger &= \sqrt{\kappa}\,\boldsymbol{Q}_2^{\dagger 1}, & \boldsymbol{q}_{\downarrow\bullet}^\dagger &= -\sqrt{\kappa}\,\boldsymbol{Q}_2^{\dagger 2},
\end{aligned}
\tag{124}
$$

for which the deformation parameters are $u = u^* = \frac{1+\kappa}{2\sqrt{\kappa}}$ and $v = v^* = \frac{1-\kappa}{2\sqrt{\kappa}}$.

The choice of whether to assign spin/charge to the bosonic/fermionic sector is not of great importance for the fundamental representation. It does however play an essential role in the identification of higher dimensional representations. As the Pauli exclusion principle limits growth of the fermionic sector, increasing $\mathcal{N}$ primarily corresponds to an increasing number of bosons. To proceed we focus on the case of assigning spin to the bosonic sector, although we highlight that the alternative possibility is also of interest as commented upon in Sec. 2.3. For the fundamental $\mathcal{N} = 1$ representation we thus adopt the identification of states

$$
|\downarrow\rangle = \boldsymbol{b}_1^\dagger |\Omega\rangle, \quad |\uparrow\rangle = \boldsymbol{b}_2^\dagger |\Omega\rangle, \quad |\circ\rangle = \boldsymbol{f}_1^\dagger |\Omega\rangle, \quad |\bullet\rangle = \boldsymbol{f}_2^\dagger |\Omega\rangle.
\tag{125}
$$

Increasing $\mathcal{N}$ then yields $4\mathcal{N}$ states which are comprised of four $su(2)$ spin-multiplets. Writing $\mathcal{N} = 2S + 1$ these are as follows: two spin-$S$ multiplets,

$$
\begin{aligned}
\frac{(\boldsymbol{b}_2^\dagger)^{S+m}(\boldsymbol{b}_1^\dagger)^{S-m}}{\sqrt{(S+m)!(S-m)!}}\boldsymbol{f}_1^\dagger |\Omega\rangle, & \qquad m \in \{-S, -S+1, \ldots, S\}, \\
\frac{(\boldsymbol{b}_2^\dagger)^{S+m}(\boldsymbol{b}_1^\dagger)^{S-m}}{\sqrt{(S+m)!(S-m)!}}\boldsymbol{f}_2^\dagger |\Omega\rangle, & \qquad m \in \{-S, -S+1, \ldots, S\},
\end{aligned}
\tag{126}
$$

a spin-$(S + \frac{1}{2})$ multiplet,

$$
\frac{(\boldsymbol{b}_2^\dagger)^{S+\frac{1}{2}+m}(\boldsymbol{b}_1^\dagger)^{S+\frac{1}{2}-m}}{\sqrt{(S+\frac{1}{2}+m)!(S+\frac{1}{2}-m)!}} |\Omega\rangle, \qquad m \in \{-S-\tfrac{1}{2}, -S+\tfrac{1}{2}, \ldots, S+\tfrac{1}{2}\},
\tag{127}
$$

and a spin-$(S - \frac{1}{2})$ multiplet,

$$
\frac{(\boldsymbol{b}_2^\dagger)^{S-\frac{1}{2}+m}(\boldsymbol{b}_1^\dagger)^{S-\frac{1}{2}-m}}{\sqrt{(S-\frac{1}{2}+m)!(S-\frac{1}{2}-m)!}}\boldsymbol{f}_1^\dagger\boldsymbol{f}_2^\dagger |\Omega\rangle, \qquad m \in \{-S+\tfrac{1}{2}, -S+\tfrac{3}{2}, \ldots, S-\tfrac{1}{2}\}.
\tag{128}
$$

This basis of states admits a natural interpretation as combining a spin-$S$ local moment with the electron. Firstly, the two spin-$S$ multiplets of Eq. (126) can be written as $|\circ; S, m\rangle$ and $|\bullet; S, m\rangle$, which are the states of a spin-moment with the electronic state respectively unoccupied and doubly occupied. The remaining two multiplets arise as $S \otimes \frac{1}{2} = (S + \frac{1}{2}) \oplus (S - \frac{1}{2})$, manifesting the entwining of the spin-moment with the electronic spin. Specifically, the the spin-$(S + \frac{1}{2})$ multiplet takes the form

$$
\gamma_{m+\frac{1}{2}}^+ |\uparrow; S, m - \tfrac{1}{2}\rangle + \gamma_{m-\frac{1}{2}}^- |\downarrow; S, m + \tfrac{1}{2}\rangle,
\tag{129}
$$

and the spin-$(S - \frac{1}{2})$ multiplet the form

$$\gamma^-_{m-\frac{1}{2}} \left|\uparrow; S, m - \tfrac{1}{2}\right\rangle - \gamma^+_{m+\frac{1}{2}} \left|\downarrow; S, m + \tfrac{1}{2}\right\rangle, \tag{130}$$

with $\gamma^\pm_m = \sqrt{\frac{S \pm m}{2S+1}}$. In this way we see that these higher dimensional representations correspond directly to the local moment DOF, as discussed in Sec. 2.3. Through this identification of basis states we obtain $\boldsymbol{q}$ of Eq. (25) from Eq. (121).

# B  Mori–Zwanzig projection scheme

In this appendix we briefly summarise the Mori–Zwanzig projection scheme [14–16], see also [58,62]. This complements the analysis of Sec. 4, allowing one to connect to the literature more broadly.

Here we formulate time evolution through the Liouville operator, $\mathcal{L}\mathcal{O} = [\boldsymbol{H}, \mathcal{O}]$, as follows

$$\frac{\mathrm{d}\boldsymbol{a}_i(t)}{\mathrm{d}t} = \mathrm{i}\mu\boldsymbol{a}_i(t) + \mathrm{i}\mathcal{L}\boldsymbol{a}_i(t), \tag{131}$$

where we have extracted explicit dependence on the chemical potential to match with the conventions of Sec. 4.1. This has the formal solution $\boldsymbol{a}(t) = e^{\mathrm{i}\mu t} e^{\mathrm{i}\mathcal{L}t} \boldsymbol{a} = e^{\mathrm{i}\mu t} \boldsymbol{a} e^{-\mathrm{i}\mathcal{L}t}$, allowing the retarded Green's function to be expressed as $G_{ij}(t) = -\mathrm{i}\vartheta(t) \langle \boldsymbol{a}_i | e^{\mathrm{i}(\mu - \mathcal{L})t} \boldsymbol{a}^\dagger_j \rangle$. Switching to frequency this gives,

$$G_{ij}(\omega) = -\mathrm{i} \int_0^\infty \mathrm{d}t \, e^{\mathrm{i}\omega t} \langle \boldsymbol{a}_i | e^{\mathrm{i}(\mu - \mathcal{L})t} \boldsymbol{a}^\dagger_j \rangle = \langle \boldsymbol{a}_i | \tfrac{1}{\omega + \mu - \mathcal{L}} \boldsymbol{a}^\dagger_j \rangle, \qquad \mathrm{Im}\,\omega > 0, \tag{132}$$

which is a formal rewriting of the equation of motion.

We proceed to project correlations onto the local DOFs by decomposing $\mathcal{L} = \mathcal{L}\boldsymbol{p} + \mathcal{L}\boldsymbol{q}$ in terms of the projection operators

$$\boldsymbol{p} = \sum_{i,j} |\boldsymbol{a}^\dagger_i\rangle \langle \boldsymbol{a}_i | \boldsymbol{a}^\dagger_j\rangle^{-1} \langle \boldsymbol{a}_j| = \sum_{i,j} |\boldsymbol{a}^\dagger_i\rangle I^{-1}_{ij} \langle \boldsymbol{a}_j|, \qquad \boldsymbol{q} = 1 - \boldsymbol{p}. \tag{133}$$

We then write

$$\frac{1}{\omega + \mu - \mathcal{L}} = \frac{1}{\omega + \mu - \mathcal{L}\boldsymbol{q}} + \frac{1}{\omega + \mu - \mathcal{L}\boldsymbol{q}} \mathcal{L}\boldsymbol{p} \frac{1}{\omega + \mu - \mathcal{L}}, \tag{134}$$

employing the operator identity $\frac{1}{\boldsymbol{a} - \boldsymbol{b}} = \frac{1}{\boldsymbol{a}} + \frac{1}{\boldsymbol{a}}\boldsymbol{b}\frac{1}{\boldsymbol{a} - \boldsymbol{b}}$, with $\boldsymbol{a} = \omega + \mu - \mathcal{L}\boldsymbol{q}$ and $\boldsymbol{b} = \mathcal{L}\boldsymbol{q}$, resulting in

$$G_{ij}(\omega) = \frac{I_{ij}}{\omega + \mu} + \sum_{k,l} \langle \boldsymbol{a}_i | \frac{1}{\omega + \mu - \mathcal{L}\boldsymbol{q}} \mathcal{L}\boldsymbol{a}^\dagger_k \rangle I_{kl} G_{lj}(\omega). \tag{135}$$

Further decomposing $\frac{1}{\omega + \mu - \mathcal{L}\boldsymbol{q}} \mathcal{L} = \frac{\mathcal{L}}{\omega + \mu} + \frac{\mathcal{L}\boldsymbol{q}}{\omega + \mu - \mathcal{L}\boldsymbol{q}} \mathcal{L}$, we arrive at

$$\sum_{k,l} \left( (\omega + \mu) I_{ik} - K^\star_{ik} - M^\star_{ik}(\omega) \right) I^{-1}_{kl} G_{lj}(\omega) = I_{ij}, \tag{136}$$

where

$$K^\star_{ij} = \langle \boldsymbol{a}_i | \mathcal{L} \boldsymbol{a}_j^\dagger \rangle = \langle \boldsymbol{a}_i \mathcal{L} | \boldsymbol{a}_j^\dagger \rangle \,, \qquad M^\star_{ij}(\omega) = \langle \boldsymbol{a}_i \mathcal{L} | \boldsymbol{q} \tfrac{1}{\omega + \mu - \boldsymbol{q} \mathcal{L} \boldsymbol{q}} \boldsymbol{q} \mathcal{L} \boldsymbol{a}_j^\dagger \rangle \,. \tag{137}$$

We see that this matches Eq. (58) upon switching to momentum space. The expressions for $K^\star_{ij}$ are identical, and consequently the expressions for $M^\star_p(\omega)$ are formally equivalent. We thus see that the Mori–Zwanzig projection scheme organises correlations in precisely the same manner as the Tserkovnikov approach of Sec. 4.1. The distinction is that here $M^\star_{ij}(\omega)$ is formulated through a static projection operator normalised by $I_{ij}$, whereas in Eq. (54) it is formulated through a dynamic projection operator normalised by $G_{ij}(\omega)$ [17]. We find this latter formulation more convenient for practical purposes, and its derivation more instructive.

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
