# Peer review of "Non-canonical degrees of freedom"

_SciPost Physics_

## Round 1 · Referee Report · Anonymous (Referee 1) · 2020-12-7

Report

The manuscript deals with the non-canonical degrees of freedom and how to construct perturbation theory for them. The latter seems to be a non-trivial question because of the modified commutation relations the Wick's theorem is not applicable, and, consequently, the diagrammatic methods are not at hand. Nevertheless, the author finds a way to formulate Dyson's equation for single-particle Green's function.
The paper is nicely written and for sure will be an invaluable reference in its field. I recommend this paper for publication in SciPost Physics.
I have several questions and remarks that can improve the presentation:
(i) in section 3 the author implies the hoppings and interactions to be symmetric; should they also be real?
(ii) For the algebra of the non-canonical DOF that renders the Heisenberg model into Eq. (31), what is the value of \lambda (\kappa)?
(iii) Is it possible to give a functional integral presentation for the non-canonical DOF? Or, in other words, is it known anything about the coherent states for the algebra Eq. (29)?
(iv) The author claims that Eqs. (75)-(82) provide a closed-form for M_p^*(\omega). I wonder if that statement could be explained a bit more in the last paragraph of section 5.2. In a sense, what are the variables, and which of Eqs. (75)-(82) are definitions, which are the relations, and which are the actual equations to solve.

  • validity: high
  • significance: high
  • originality: high
  • clarity: high
  • formatting: excellent
  • grammar: excellent

Author:  Eoin Quinn  on 2021-02-01  [id 1195]

(in reply to Report 1 on 2020-12-07)

We are grateful to the Referee for their assessment of our manuscript and for their remarks. Let us address each point in turn.

(i) in section 3 the author implies the hoppings and interactions to be symmetric; should they also be real?

Yes for a Hermitian Hamiltonian these parameters should be real if they are symmetric. We have modified the text to clarify that we take them to be real.

(ii) For the algebra of the non-canonical DOF that renders the Heisenberg model into Eq. (31), what is the value of \lambda (\kappa)?

We have modified the text to specify that here \lambda = 1/S.

(iii) Is it possible to give a functional integral presentation for the non-canonical DOF? Or, in other words, is it known anything about the coherent states for the algebra Eq. (29)?

In general we do not know an answer to this. While there exist textbook studies on coherent states for spin systems, we have been unable to find concrete results on coherent states for the non-canonical formulation of the electron, i.e. for Hubbard operators. We agree with the referee that this would be interesting to explore, and while we do not feel that we can add constructive comments at this point, we hope that this can be addressed in future work.

(iv) The author claims that Eqs. (75)-(82) provide a closed-form for M_p^*(\omega). I wonder if that statement could be explained a bit more in the last paragraph of section 5.2. In a sense, what are the variables, and which of Eqs. (75)-(82) are definitions, which are the relations, and which are the actual equations to solve.

We thank the Referee for raising this. We have now split this paragraph in two to first focus on the form of the closed expression for M_p^*(\omega). We now clearly point to the closed set of equations, framed in terms of the unknown G. We have also modified the text to clarify that the closed equations take a functional differential form, whose solution requires a perturbative expansion. To highlight that this is analogous to the corresponding canonical case we have added a reference for the corresponding closed equation for the canonical self-energy to paragraph 3 of the Introduction.

---

## Round 1 · Referee Report · Anonymous (Referee 3) · 2021-1-22

Report

The revised manuscript presents a detailed description of non-canonical degrees of freedom, and generalized Dyson equation, RPA technique etc for non-canonical degrees of freedom.
Since most of the text books currently discuss only canonical degrees of freedom, though there are many important topics in condensed matter physics and beyond, where non-canonical degrees of freedom might play crucial role, I think this work will provide a good reference in future.
I think the revised manuscript satisfies the criterion of SciPost Core and I recommend it for publication.

---

## Round 1 · Referee Report · Anonymous (Referee 2) · 2021-1-22

Report

The revised manuscript presents a detailed description of non-canonical degrees of freedom, and generalized Dyson equation, RPA technique etc for non-canonical degrees of freedom.
Since most of the text books currently discuss only canonical degrees of freedom, though there are many important topics in condensed matter physics and beyond, where non-canonical degrees of freedom might play crucial role, I think this work will provide a good reference in future.
I think the revised manuscript satisfies the criterion of SciPost Core and I recommend it for publication.

  • validity: -
  • significance: -
  • originality: -
  • clarity: -
  • formatting: -
  • grammar: -

Author:  Eoin Quinn  on 2021-02-01  [id 1194]

(in reply to Report 2 on 2021-01-22)
Category:
remark

We thank the Referee for their report.

---

## Editorial Decision

resubmitted